# $^{18}$F-FDG-PET/MR imaging to monitor disease activity in large vessel vasculitis

Dan Pugh[1], Dilip Patel[2], Gillian Macnaught[3], Alicja Czopek[1], Lorraine Bruce[1], James Donachie[1], Peter J. Gallacher[1], Sovira Tan[4], Mark Ahlman[5], Peter C. Grayson[4], Neil Basu[6] & Neeraj Dhaun [1] ✉

Disease-monitoring in large vessel vasculitis (LVV) is challenging. Simultaneous $^{18}$F-fluorodeoxyglucose positron emission tomography with magnetic resonance imaging (PET/MRI) provides functional assessment of vascular inflammation alongside high-definition structural imaging with a relatively low burden of radiation exposure. Here, we investigate the ability of PET/MRI to monitor LVV disease activity longitudinally in a prospective cohort of patients with active LVV. We demonstrate that both the PET and MRI components of the scan can distinguish active from inactive disease using established quantification methods. Using logistic-regression modelling of PET/MRI metrics, we devise a novel PET/MRI-specific **V**asculitis **A**ctivity using **M**R **P**ET (**VAMP**) score which is able to distinguish active from inactive disease with more accuracy than established methods and detects changes in disease activity longitudinally. These findings are evaluated in an independent validation cohort. Finally, PET/MRI improves clinicians' assessment of LVV disease activity and confidence in disease management, as assessed via clinician survey. In summary, PET/MRI may be useful in tracking disease activity and assessing treatment-response in LVV. Based on our findings, larger, prospective studies assessing PET/MRI in LVV are now warranted.

Large vessel vasculitis (LVV), incorporating giant cell arteritis (GCA) and Takayasu arteritis (TAK), is the most common primary vasculitis and is characterised by chronic inflammation of medium and large arteries[1,2]. Arterial narrowing may lead to limb ischaemia or organ dysfunction. Alongside this, risk of aortic aneurysm formation and rupture is also increased[3]. Although effective treatments are available[4], most patients with LVV suffer a relapsing and remitting course[5,6]. Monitoring disease activity in LVV is challenging as symptoms and clinical signs are non-specific. Moreover, patients may have active and progressive disease whilst receiving treatment, despite absent clinical features and suppressed serological markers of inflammation[7]. This leaves patients at risk of over-treatment, including the adverse effects

of glucocorticoids, or under-treatment, leading to unchecked vascular inflammation and damage. The lack of a robust disease monitoring tool has also limited the design of clinical trials in LVV.

Vascular imaging is a key diagnostic tool in LVV[8]. However, no currently available imaging modality can reliably monitor disease activity and treatment response throughout the disease course[9]. Reasons for this include lack of sufficient coverage of the vasculature, an inability to distinguish active inflammation from arterial remodelling or atherosclerosis (a process that is accelerated in LVV[10,11]), and unacceptable radiation exposure. Thus, there is an urgent, unmet need for an imaging modality that can accurately and safely monitor disease activity longitudinally[8]. Such an imaging modality would be an

[1]BHF/University Centre for Cardiovascular Science, University of Edinburgh, Edinburgh, UK. [2]Department of Radiology, Royal Infirmary of Edinburgh, Edinburgh, UK. [3]Edinburgh Imaging Facility, University of Edinburgh, Edinburgh, UK. [4]National Institute of Arthritis & Musculoskeletal & Skin Diseases, National Institutes of Health, Bethesda, MD, USA. [5]Department of Radiology & Imaging, Medical College of Georgia, Georgia, USA. [6]Institute of Infection, Immunity & Inflammation, University of Glasgow, Glasgow, UK. ✉e-mail: bean.dhaun@ed.ac.uk

invaluable research tool for evaluating novel treatments for LVV, an important consideration given the large number of emerging candidate therapies.

Hybrid positron emission tomography with magnetic resonance imaging (PET/MRI) has emerged as a powerful imaging modality in cancer[12] and cardiovascular disease[13,14]. It combines quantification of metabolic activity using PET with the superior soft tissue definition of MRI, thus providing a more detailed assessment of vessel wall morphology, potentially allowing discrimination between atheroma and inflammation[15,16]. In addition to high-resolution anatomical assessment, MRI provides a functional assessment of vascular inflammation which complements that provided by PET[17,18]. Importantly, PET/MRI requires significantly less radiation exposure than PET combined with computed tomography (PET/CT), allowing interval imaging, even in young patients. Although two small, cross-sectional studies have explored the potential role of PET/MRI in the diagnosis of LVV[19,20], no study has robustly examined its ability to monitor disease activity longitudinally, or defined a clinically useful method to evaluate the images. Here, in a prospective, longitudinal cohort of patients with new or relapsing LVV, we evaluated the ability of PET/MRI to determine arterial inflammation and its response to treatment. Additionally, we developed and validated a novel, clinically useful, PET/MRI-specific scoring system for quantifying disease activity in LVV.

## Results
### Study population characteristics
Twenty-four subjects were recruited and underwent a total of 40 PET/MRI scans (Supplementary fig. 1). Fourteen had a diagnosis of GCA, 6 had a diagnosis of TAK and 4 had LVV not otherwise specified. The mean age of subjects at baseline was $61 \pm 15$ years and 17 (71%) were female (Table 1). Based on clinical assessment, 17 of 24 patients (71%) had active LVV at the time of baseline PET/MR imaging and 6/16 (38%) had active disease at follow-up. All scans were completed suggesting good tolerability. No adverse events were encountered.

### Overall assessment of PET/MR imaging
There was good agreement between clinician assessment of LVV disease activity and blinded radiologist interpretation of disease activity based on PET/MR imaging alone (Supplementary table 1). Considering both baseline and follow-up imaging, the radiologist's interpretation alone had a sensitivity of 78% and specificity of 88% for distinguishing active from inactive LVV.

### Assessment of PET imaging
1. Visual comparison with liver uptake: $^{18}$F-fluorodeoxyglucose (FDG) uptake was greater in active versus inactive disease in all aortic territories and in the brachiocephalic and subclavian arteries (active vs. inactive disease: ascending aorta $2.0 \pm 0.8$ vs. $1.1 \pm 0.6$, $P = 0.003$; aortic arch $2.1 \pm 0.8$ vs. $1.3 \pm 0.7$, $P = 0.02$; descending thoracic aorta $2.2 \pm 0.8$ vs. $1.4 \pm 0.8$, $P = 0.03$; abdominal aorta $2.1 \pm 0.8$ vs. $1.4 \pm 0.8$, $P = 0.03$; brachiocephalic artery $1.5 \pm 1.0$ vs. $0.8 \pm 0.8$, $P = 0.03$; subclavian arteries $1.2 \pm 1.1$ vs. $0.5 \pm 0.5$, $P = 0.04$). No significant differences were observed between active and inactive disease in the remaining arterial territories (Fig. 1). Uptake was also greater in GCA versus TAK, most notably in the carotid arteries ($1.7 \pm 0.8$ vs. $0.7 \pm 0.7$, $P = 0.03$; Supplementary fig. 2).

2. PET Vasculitis Activity Score (PETVAS): PETVAS was higher in patients with active disease compared to those with inactive disease ($15.6 \pm 7.0$ vs. $8.8 \pm 4.2$, $P = 0.001$; Fig. 2A). PETVAS was also higher in patients with GCA compared with TAK ($14.9 \pm 7.0$ vs. $9.3 \pm 4.0$, $P = 0.02$), and in those with active GCA versus active TAK ($18.2 \pm 6.1$ vs. $9.7 \pm 4.8$, $P = 0.004$; Fig. 2B). Receiver operating characteristic (ROC) analysis demonstrated an area under the curve (AUC) for PETVAS of 0.79 (95%CI 0.64-0.93, $P = 0.002$) with a suggested cut-off value of 12

to distinguish active from inactive disease (Fig. 2A). Using this threshold, PETVAS demonstrated a sensitivity of 74% (95%CI 54-87%) and specificity of 76% (95%CI 53-90%) for distinguishing active from inactive disease, modestly inferior to a blinded radiologist's assessment.

3. Maximum standardised uptake value ($SUV_{max}$) target-to-background (TBR) scoring: Using $SUV_{max}$ TBR scoring, greater FDG uptake was observed in the ascending thoracic aorta and abdominal aorta in patients with active disease versus inactive disease (Fig. 3A). There was no difference between those with GCA and TAK (Supplementary fig. 3).

4. $SUV_{mean}$ TBR scoring: $SUV_{mean}$ TBR scoring demonstrated increased uptake in active versus inactive disease in all four aortic territories as well as the right carotid artery and left subclavian artery (Fig. 3B). This compared favourably with established quantification methods, including $SUV_{max}$ TBR. A cumulative $SUV_{mean}$ TBR score, combining scores from all 12 arterial territories, was higher in patients with active disease compared to those with inactive disease ($15.6 \pm 2.8$ vs. $12.9 \pm 0.8$, $P = 0.0004$; Fig. 3C).

### Assessment of MR imaging
Mural thickness: There was no difference in maximum aortic wall thickness between patients with clinically determined active and inactive disease ($4.3 \pm 1.3$ mm vs. $4.4 \pm 1.4$ mm, $P = 0.9$; Fig. 4A).

T2-weighted mural signal: the number of arterial territories with increased T2-weighted mural signal was greater in subjects with active versus inactive LVV ($2.4 \pm 3.3$ vs. $0.1 \pm 0.5$, $P = 0.007$; Fig. 4B). Similarly, the proportion of scans which demonstrated any evidence of increased T2-weighted mural signal was greater in active versus inactive disease in all arterial territories except iliac arteries.

Mural enhancement: the number of arterial territories with mural enhancement was no different in patients with active LVV versus those with inactive LVV ($1.8 \pm 2.3$ vs. $0.6 \pm 1.0$, $P = 0.05$; Fig. 4C).

Luminal abnormalities: the number of luminal abnormalities observed (stenosis, occlusion, dilation or aneurysm) ranged from 0 to 7 and did not differ between those with active and inactive disease (Fig. 4D and Supplementary fig. 4). However, more luminal abnormalities were observed in those with TAK compared to those with GCA (Fig. 4E).

### Longitudinal assessment
Fourteen patients had >1 PET/MRI scan with a median of 217 [187-287] days between the first and final scan. PETVAS fell significantly from baseline to follow-up ($18.2 \pm 6.4$ vs. $9.6 \pm 4.9$, $P = 0.0004$; Fig. 5A). Including only those patients who were considered to have clinically active disease at baseline and inactive disease at follow-up, PETVAS fell from $20.5 \pm 5.2$ to $8.9 \pm 3.7$ ($P = 0.0007$; Fig. 5B).

Using $SUV_{max}$ TBR and including all patients, a change from baseline to follow-up was observed in the descending thoracic aorta only ($2.6 \pm 0.9$ vs. $2.1 \pm 0.6$, $P = 0.008$; Fig. 5C). Including only those patients who were considered to have clinically active disease at baseline and inactive disease at follow-up, $SUV_{max}$ TBR fell in the ascending aorta ($2.3 \pm 0.6$ vs. $1.8 \pm 0.3$, $P = 0.02$), descending aorta ($2.7 \pm 1.0$ vs. $2.0 \pm 0.6$, $P = 0.02$) and abdominal aorta ($2.6 \pm 0.8$ vs. $1.9 \pm 0.4$, $P = 0.04$; Fig. 5D).

Using $SUV_{mean}$ TBR, a change from baseline to follow-up was observed across four different arterial territories; descending aorta ($1.5 \pm 0.3$ vs. $1.2 \pm 0.1$, $P = 0.005$), right carotid ($1.6 \pm 0.4$ vs. $1.3 \pm 0.2$, $P = 0.01$), left carotid ($1.5 \pm 0.3$ vs. $1.2 \pm 0.1$, $P = 0.03$) and right subclavian ($1.3 \pm 0.3$ vs. $1.0 \pm 0.2$, $P = 0.002$) arteries. Including only those patients who were considered to have clinically active disease at baseline and inactive disease at follow-up, $SUV_{mean}$ TBR fell in all aortic territories as well as right and left carotid arteries and right and left subclavian arteries (ascending aorta: $1.4 \pm 0.3$ vs. $1.1 \pm 0.1$, $P = 0.008$; aortic arch: $1.4 \pm 0.2$ vs. $1.1 \pm 0.1$, $P = 0.008$; descending aorta: $1.4 \pm 0.3$

**Table 1 | Baseline subject characteristics**

| Parameter | Baseline | | | Follow-up | | |
|---|---|---|---|---|---|---|
| | Active disease based on clinician assessment | Inactive disease based on clinician assessment | Total | Active disease based on clinician assessment | Inactive disease based on clinician assessment | Total |
| n | 17 | 7 | 24 | 6 | 10 | 16 |
| Age, years | 57 ± 15 | 70 ± 10 | 61 ± 15 | 54 ± 21 | 63 ± 12 | 60 ± 16 |
| Female (%) | 13 (76) | 4 (57) | 17 (71) | 5 (83) | 7 (70) | 12 (75) |
| White (%) | 17 (100) | 7 (100) | 24 (100) | 6 (100) | 10 (100) | 16 (100) |
| Smoking (%) | | | | | | |
| Current | 4 (24) | 0 (0) | 4 (17) | 1 (17) | 1 (10) | 2 (13) |
| Ex- | 3 (18) | 4 (57) | 7 (29) | 0 (0) | 2 (20) | 2 (13) |
| Never | 10 (59) | 3 (43) | 13 (54) | 5 (83) | 7 (70) | 12 (75) |
| Diabetes (%) | 0 (0) | 0 (0) | 0 (0) | 0 (0) | 0 (0) | 0 (0) |
| Hypertension (%) | 2 (12) | 5 (71) | 7 (29) | 0 (0) | 2 (20) | 2 (13) |
| **Disease characteristics** | | | | | | |
| Diagnosis (%) | | | | | | |
| GCA | 13 (76) | 1 (14) | 14 (58) | 3 (50) | 9 (90) | 12 (75) |
| TAK | 4 (24) | 2 (29) | 6 (25) | 3 (50) | 1 (10) | 4 (25) |
| Unspecified | 0 (0) | 4 (57) | 4 (17) | 0 (0) | 0 (0) | 0 |
| Disease duration, days | 100 [39–286] | 88 [9–986] | 94 [38–360] | 444 [314–711] | 397 [225–572] | 418 [252–543] |
| Immunosuppression at time of PET/MRI (%) | | | | | | |
| Prednisolone | 9 (53) | 3 (43) | 12 (50) | 4 (67) | 10 (100) | 14 (88) |
| Methotrexate | 1 (6) | 1 (14) | 3 (13) | 2 (33) | 2 (20) | 4 (25) |
| MMF | 1 (6) | 0 (0) | 1 (4) | 1 (17) | 1 (10) | 2 (13) |
| Tocilizumab | 0 (0) | 0 (0) | 0 | 4 (67) | 2 (20) | 7 (44) |
| Nil | 7 (41) | 4 (57) | 11 (46) | 0 (0) | 0 (0) | 0 |
| Duration of immunosuppression, days | 5 [0–18] | 0 [0–420] | 4 [0–19] | 190 [163–383] | 213 [180–390] | 205 [180–385] |
| **Clinical** | | | | | | |
| Systolic BP, mmHg | 137 ± 19 | 140 ± 21 | 138 ± 19 | 154 ± 33 | 145 ± 23 | 148 ± 27 |
| Diastolic BP, mmHg | 80 ± 14 | 75 ± 9 | 79 ± 13 | 85 ± 19 | 80 ± 16 | 82 ± 17 |
| Heart rate, bpm | 83 ± 11 | 86 ± 21 | 84 ± 14 | 78 ± 14 | 73 ± 15 | 75 ± 14 |
| Body mass index, kg/m$^2$ | 23.9 ± 3.9 | 28.3 ± 4.7 | 25.2 ± 4.5 | 24.1 ± 2.4 | 25.2 ± 3.5 | 25.1 ± 3.0 |
| **Laboratory** | | | | | | |
| Haemoglobin, g/L | 123 ± 15 | 128 ± 29 | 125 ± 18 | 133 ± 16 | 141 ± 15 | 138 ± 15 |
| Leucocytes, x10$^9$/L | 10.9 ± 3.5 | 10.4 ± 3.6 | 10.7 ± 3.5 | 10.1 ± 5.1 | 8.8 ± 2.5 | 9.3 ± 3.5 |
| Platelets, x10$^9$/L | 417 ± 133 | 416 ± 223 | 417 ± 159 | 320 ± 60 | 307 ± 116 | 311 ± 98 |
| C-reactive protein, mg/L | 18 [2–66] | 11 [3–69] | 13 [3–63] | 1 [0-7] | 2 [1–12] | 2 [0-10] |
| ESR, mm/h | 53 ± 36 | 43 ± 41 | 49 ± 37 | 13 ± 9 | 10 ± 6 | 11 ± 7 |
| Creatinine, mg/dL | 0.70 ± 0.12 | 0.93 ± 0.20 | 0.77 ± 0.18 | 0.79 ± 0.16 | 0.75 ± 0.06 | 0.78 ± 0.12 |
| LRG1, ng/mL | 72.2 ± 57.7 | 61.8 ± 28.3 | 69.5 ± 51.3 | 39.2 ± 18.6 | 35.7 ± 14.3 | 36.8 ± 15.2 |
| Ang-2, pg/mL | 3,443 ± 1,714 | 3,107 ± 1,512 | 3,365 ± 1,646 | 2,503 ± 896 | 1,996 ± 597 | 2,154 ± 715 |

Data are presented as (%), mean ± SD, or median [IQR].

*Ang-2* angiopoietin-2, *BP* blood pressure, *ESR* erythrocyte sedimentation rate, *GCA* giant cell arteritis, *LRG1* leucine-rich α-2 glycoprotein 1, *MMF* mycophenolate mofetil, *TAK* Takayasu arteritis.

vs. 1.1 ± 0.1, $P = 0.03$; abdominal aorta: 1.3 ± 0.3 vs. 1.1 ± 0.1, $P = 0.02$; right carotid: 1.6 ± 0.1 vs. 1.3 ± 0.1, $P = 0.003$; left carotid: 1.5 ± 0.2 vs. 1.2 ± 0.1, $P = 0.008$; right subclavian: 1.3 ± 0.3 vs. 1.1 ± 0.2, $P = 0.02$; left subclavian: 1.4 ± 0.3 vs. 1.1 ± 0.2, $P = 0.002$).

Cumulative SUV$_{mean}$ TBR score also fell significantly from baseline to follow-up (4.1 ± 3.1 vs. 1.6 ± 1.9, $P = 0.02$). Including only those patients who were considered to have clinically active disease at baseline and inactive disease at follow-up, cumulative SUV$_{mean}$ TBR score fell from 4.3 ± 2.1 to 1.2 ± 0.8 ($P = 0.005$).

T2-weighted mural signal fell significantly in those with active disease at baseline and inactive disease at follow-up (number of territories with increased mural signal: 4.1 ± 3.5 vs. 0.0 ± 0.0, $P = 0.01$; Fig. 5E). The remaining MRI metrics (mural thickness, mural enhancement and number of luminal abnormalities) did not change significantly between baseline and follow-up.

### Development of a novel PET/MRI scoring system for LVV

Using univariable logistic regression of PET metrics, SUV$_{mean}$ TBR was associated most strongly with active LVV with a positive predictive value (PPV) of 90% and negative predictive value (NPV) of 79%. Total SUV$_{mean}$ TBR score, which included both aortic and great vessel combined scores, was more powerful than either score alone. PETVAS outperformed SUV$_{max}$ TBR, which was the metric least strongly associated with active disease (Supplementary table 2).

With respect to MRI metrics, both the presence of increased mural signal on T2-weighted imaging (PPV 90%, NPV 55%) and the total

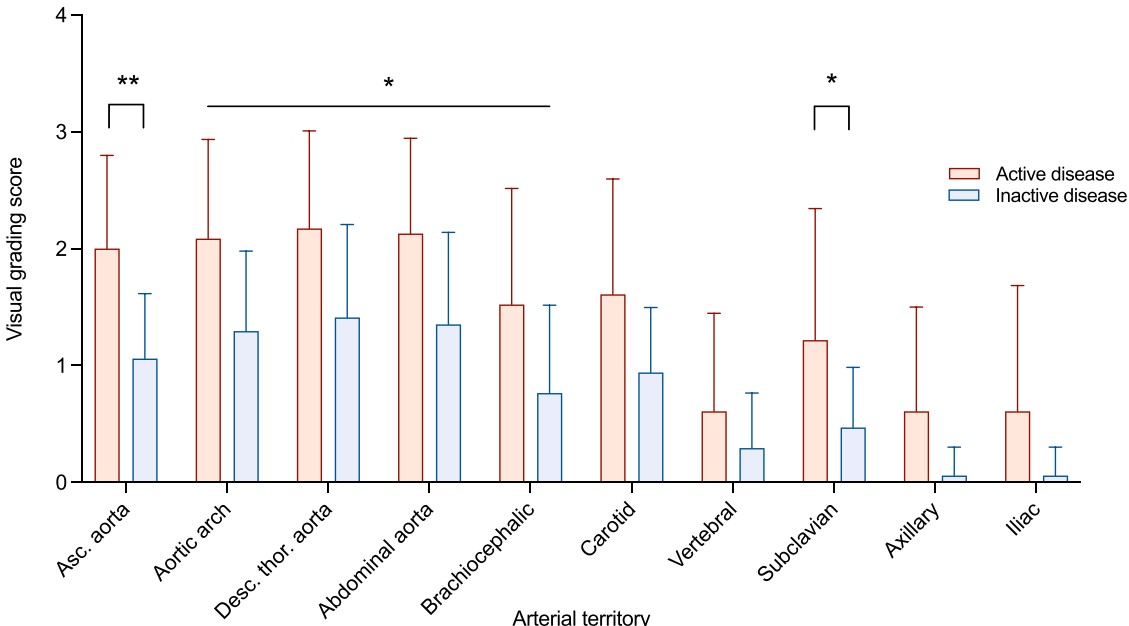

**Fig. 1 | Visual assessment of PET/MRI disease activity in LVV.** Degree of FDG uptake based on visual comparison between arterial territories of interest and the liver in those with clinically determined active (red, $n = 23$) and inactive (blue, $n = 17$) disease. FDG uptake was greater in active vs. inactive disease in all aortic territories (ascending aorta, $P = 0.003$; aortic arch, $P = 0.02$; descending thoracic aorta, $P = 0.03$; abdominal aorta, $P = 0.03$), brachiocephalic artery ($P = 0.03$) and subclavian arteries ($P = 0.04$). Data are presented as mean values ±SD. Analysis by two-way ANOVA with Šidák's multiple comparison test. Asc. aorta ascending aorta, desc. thor. aorta descending thoracic aorta.

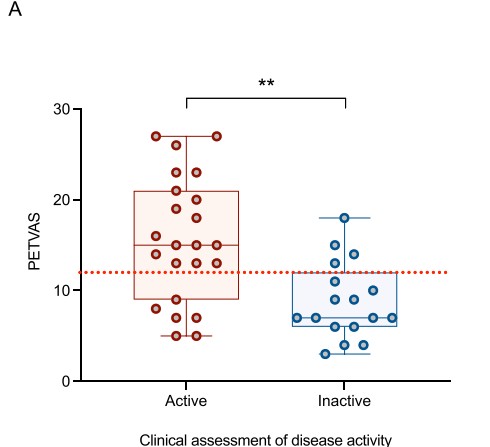

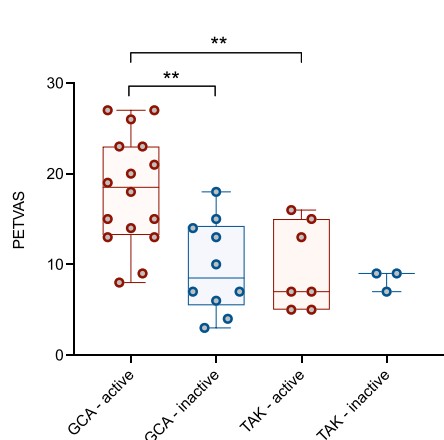

**Fig. 2 | Using PETVAS (derived from PET/MRI) to assess disease activity in LVV.** **A** Comparison of PETVAS in those with clinically determined active (red, $n = 23$) and inactive (blue, $n = 17$) disease ($P = 0.0009$). Red line denotes ROC analysis cut-off for active versus inactive disease. **B** Comparison of PETVAS in those with clinically determined active and inactive disease, further stratified by LVV subtype (GCA active vs. inactive, $P = 0.001$; GCA active vs. TAK active, $P = 0.004$). For box and whisker plots, the central line represents the median, the box represents the interquartile range, and the whiskers represent minimum and maximum values. Analysis by two-sided unpaired t-test. GCA giant cell arteritis, PETVAS PET Vasculitis Activity Score, TAK Takayasu arteritis.

number of arterial segments with increased mural signal on T2-weighted imaging (PPV 91%, NPV 57%) associated most strongly with active LVV. The remaining MRI metrics (mural enhancement, mural thickness, presence of luminal abnormalities) did not distinguish active from inactive disease (Supplementary table 3).

PET and MRI metrics that associated most strongly with active LVV in univariable analyses were then incorporated into a multivariable logistic regression analysis (Supplementary table 4). Given the relatively subjective nature of T2-weighted mural signal assessment, this metric was included in a dichotomized form (i.e., absent or present) in order to improve accuracy. The resulting analysis demonstrated that a

model including total $SUV_{mean}$ TBR score and total arterial segments with increased T2-weighted mural signal associated better with the presence of active LVV than either metric in isolation. Other PET and MRI metrics (with $P$ values < 0.05 from univariable analysis) added sequentially to the model did not improve its power.

Informed by logistic regression modelling, $SUV_{mean}$ TBR and the presence of increased mural signal on T2-weighted MRI were incorporated into a PET/MRI-specific disease activity scoring system for use in LVV. As $SUV_{mean}$ values for each arterial territory were divided by a neutral background value to create $SUV_{mean}$ TBR scores, the theoretical minimum $SUV_{mean}$ TBR score for each territory was 1 (e.g.,

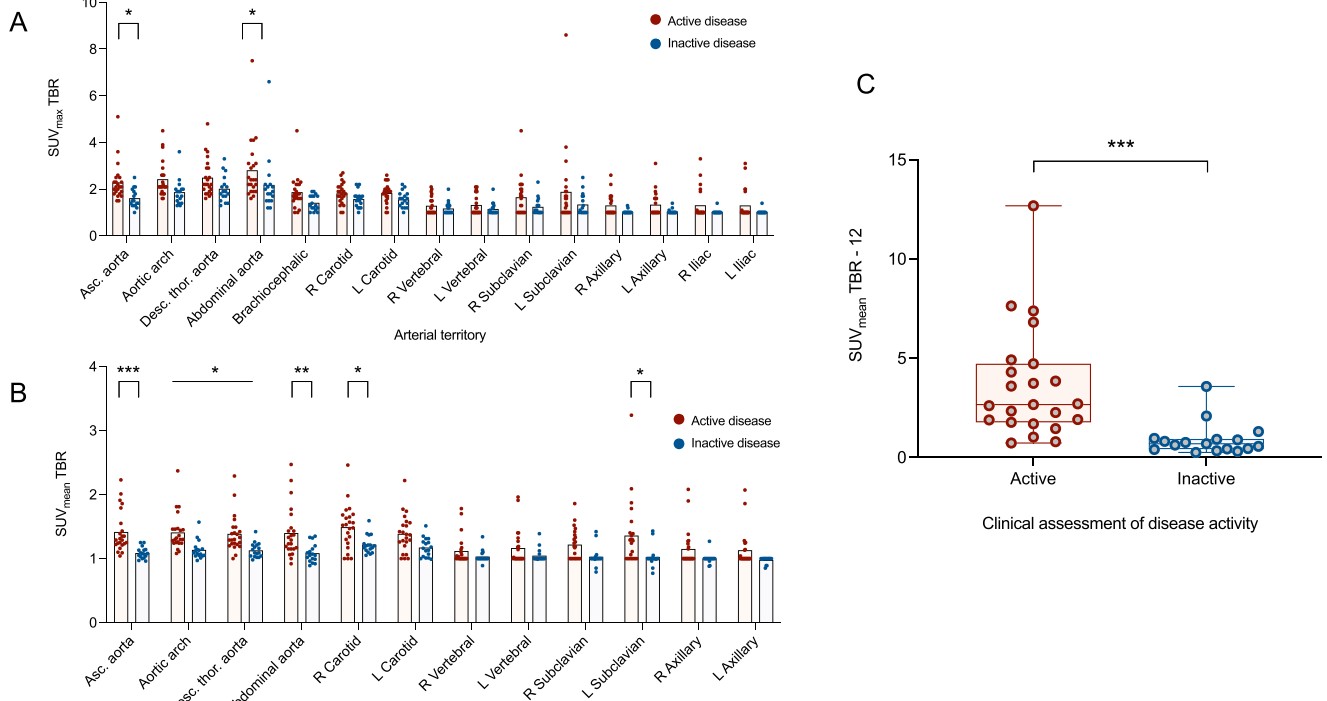

**Fig. 3 | Using SUV$_{max}$ and SUV$_{mean}$ (derived from PET/MRI) to assess disease activity in LVV. A** Degree of FDG uptake based on SUV$_{max}$ TBR analysis in those with clinically determined active and inactive disease. FDG uptake was greater in active vs. inactive disease in the ascending aorta ($P = 0.03$) and abdominal aorta ($P = 0.04$) only. Data are presented as mean values + individual data points. Analysis by two-way ANOVA with Šidák's multiple comparison test. **B** SUV$_{mean}$ TBR values allowed distinction between active and inactive disease in all aortic territories (ascending aorta, $P = 0.0009$; aortic arch, $P = 0.01$; descending thoracic aorta, $P = 0.02$; abdominal aorta, $P = 0.002$), plus right carotid ($P = 0.01$) and left sub-clavian ($P = 0.0007$). Data are presented as mean values + individual data points.

Analysis by two-way ANOVA with Šidák's multiple comparison test. **C** Comparison of cumulative SUV$_{mean}$ TBR scores in those with clinically determined active and inactive disease ($P = 0.0004$). Red dots represent patients with clinically active disease ($n = 23$), blue dots represent clinically inactive disease ($n = 17$). For box and whisker plots, the central line represents the median, the box represents the interquartile range, and the whiskers represent minimum and maximum values. Analysis by two-sided Mann–Whitney test. Asc. aorta ascending aorta, desc. thor. aorta descending thoracic aorta, L left, R right, SUVmax maximum standardised uptake value, TBR target to background ratio.

SUV$_{mean}$ of arterial territory exactly equal to SUV$_{mean}$ of background, indicating no disease activity). As 12 arterial territories were included, 12 was subtracted from the total SUV$_{mean}$ TBR score to simplify the score. Thus, if all arterial territories were equivalent to bloodpool (*i.e.*, no disease activity at all) the total SUV$_{mean}$ TBR score would equal 0. Using this method, the mean total SUV$_{mean}$ TBR score for our cohort was 2.7. ORs were then used to calculate the relative weighting of SUV$_{mean}$ TBR and the presence of increased T2-weighted mural signal. This weighting was then applied to create the final scoring system, the **V**asculitis **A**ctivity using **M**R and **P**ET (**VAMP**) score, which was calculated as follows:

VAMP = (Sum of SUV$_{mean}$ TBRs for all territories + 0.5 if aortic T2-weighted mural signal increased + 0.5 if great vessel T2-weighted mural signal increased) −12

**VAMP score – assessment of intra- and inter-operator reliability**
Intra-operator reliability was 0.99 (95%CI 0.98-1.00), as assessed by Pearson's correlation coefficient. Inter-operator reliability was 0.96 (95%CI 0.91-0.98). Bland-Altman analysis also demonstrated high intra- and inter-operator reliability (Supplementary fig. 5).

**Application of the VAMP score**
The VAMP score reliably discriminated active from inactive disease in our LVV cohort (3.9 ± 3.1 vs. 0.9 ± 0.8, $P < 0.0001$; Fig. 6A). ROC analysis demonstrated an AUC of 0.90 ($P < 0.0001$) with a suggested cut-off of 0.985 (rounded up to 1) – sensitivity 96% (95%CI 79-99), specificity 82% (95%CI 78-99) – to separate active from inactive disease. This compared favourably to PETVAS (AUC 0.79, sensitivity 74%, specificity 76%). There

was a strong correlation between VAMP score and clinical assessment of disease activity on a 0-10 scale ($r = 0.68$, $P ≤ 0.0001$; Fig. 6B). The ability of the VAMP score to distinguish active from inactive disease was stronger in GCA (4.5 ± 3.2 vs. 1.2 ± 1.0, $P = 0.003$) than TAK (2.6 ± 2.7 vs. 0.9 ± 0.8, $P = 0.3$; Supplementary fig. 6). No difference in VAMP score was observed between those with new onset versus relapsing disease (5.0 ± 3.5 vs. 2.5 ± 1.2, $P = 0.05$).

When applied to the longitudinal cohort as a whole, VAMP score fell significantly from baseline to follow-up (4.6 ± 3.4 vs. 1.7 ± 2.1, $P = 0.003$; Fig. 6C). Including only those patients who were considered to have clinically active disease at baseline and inactive disease at follow-up ($n = 8$) VAMP score fell from 4.2 ± 1.8 to 0.9 ± 0.4 ($P = 0.002$; Figs. 6D and 7).

**Clinical correlations**
We observed associations between both the PETVAS and VAMP score and clinical indicators of LVV disease activity, including C-reactive protein (CRP) and erythrocyte sedimentation rate (ESR) as well as the emerging biomarkers leucine-rich α-2 glycoprotein 1 (LRG1), angiopoietin-2 (Ang-2), calprotectin and osteopontin (Supplementary fig. 7). These associations were stronger for the VAMP score than for PETVAS (Table 2).

Interestingly, in patients with clinically active disease at the time of scanning and who had received ≥1 week of high-dose gluco-corticoids, we found that both PETVAS and VAMP score were no different, but CRP was significantly lower (<10 mg/dL in >80% of patients), compared to those patients who had received <1 week of high-dose glucocorticoids (Supplementary fig. 8).

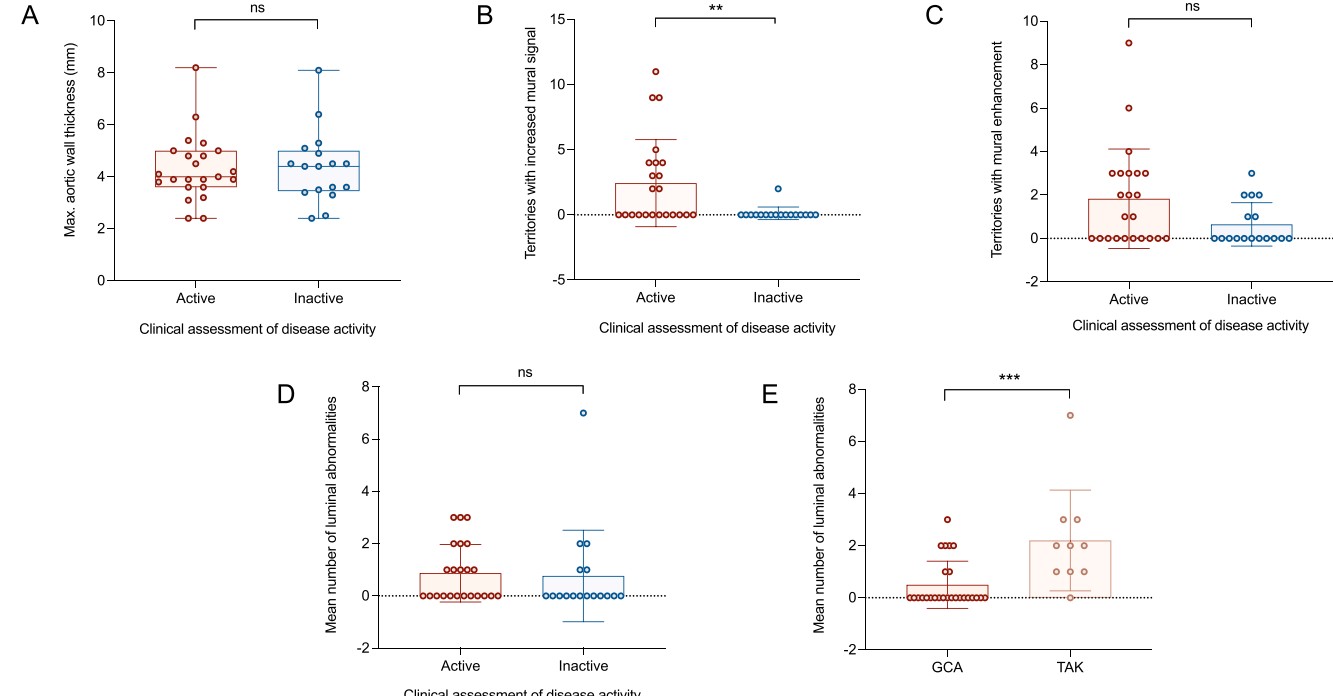

**Fig. 4 | Using MRI metrics to assess disease activity in LVV. A** Comparison of aortic wall thickness in those with clinically determined active and inactive disease. **B** Comparison of arterial territories with increased mural signal on T2-weighted MR images in those with clinically determined active and inactive disease (*P* = 0.007). **C** Comparison of arterial territories with mural enhancement on post-contrast T1-weighted MR images in those with clinically determined active and inactive disease. **D** Comparison of number of luminal abnormalities in those with clinically determined active and inactive disease. For **A**–**D**, red dots represent patients with clinically active disease (*n* = 23), blue dots represent clinically inactive disease (*n* = 17). **E** Comparison of number of luminal abnormalities in those with GCA (*n* = 26) and TAK (*n* = 10; *P* = 0.0009). For box and whisker plots, the central line represents the median, the box represents the interquartile range, and the whiskers represent minimum and maximum values. Analysis by two-sided unpaired/paired t-test. GCA giant cell arteritis, TAK Takayasu arteritis.

## Validation of the VAMP score

In an independent validation cohort of 64 PET/MRI scans obtained from patients with LVV, correlation was observed between clinical assessment of disease activity (using physician's global assessment (PGA)) and the VAMP score (*r* = 0.26, *P* = 0.04). This correlation was driven by those patients with GCA (*r* = 0.34, *P* = 0.03, *n* = 42) as opposed to those with TAK (*r* = −0.14, *P* = 0.5, *n* = 22). No correlation was observed between clinical assessment and PETVAS in the overall cohort (*r* = 0.22, *P* = 0.08; Supplementary fig. 9).

## Assessing the clinical utility of PET/MRI in LVV

Finally, we determined the potential clinical utility of PET/MRI in the management of patients with LVV. Questionnaires including clinical vignettes of patients with active/inactive LVV were sent to 18 clinicians experienced in the management of LVV. These vignettes asked clinicians to provide their opinions on disease activity and patient management. Scenarios initially excluded data from PET/MRI and then the same questions were asked with PET/MRI data included. Ten fully completed questionnaires (56%) were returned. Disease activity assessment was highly variable and improved significantly following disclosure of PET/MRI results (correct identification of active disease: 30% pre-PET/MRI vs. 90% post-PET/MR, *P* = 0.003; correct identification of inactive disease: 52% pre-PET/MRI vs. 84% post-PET/MRI, *P* = 0.07). Clinicians' confidence scores for decision making regarding disease activity and altering immunosuppression also improved, from 73/100 to 84/100 (*P* = 0.0007), and from 77/100 to 86/100 (*P* = 0.008), respectively (Fig. 8).

## Discussion

Objective measures of disease activity in LVV are lacking and this continues to impact patient care. As the number of targeted therapies

for LVV increases, a biomarker capable of reliably informing a disease-responsive approach to management is urgently needed and would be expected to improve patient outcomes. Our study suggests that PET/MRI, which provides whole-body high-definition anatomical data and a functional assessment of vascular inflammation, may be fit for this purpose. In this prospective study, we have shown that PET/MRI could distinguish active from inactive disease and track disease activity over time. Additionally, we have developed a novel, PET/MRI-specific scoring system – the VAMP score – which out-performed blinded clinician-experts and established scores at distinguishing active from inactive LVV, reflected treatment-response and which we validated in an independent patient cohort. Finally, PET/MRI improved clinicians' decision making with respect to patient care.

PET/CT is a commonly applied imaging modality currently used to assess disease activity in patients with LVV[2]. However, it has limitations. In their 2018 study, Grayson et al. demonstrated that whereas PET/CT was good at discriminating active LVV from non-vasculitis comparators using radiologist's interpretation alone, it was poor at distinguishing active from inactive LVV. This improved (sensitivity to 68%; specificity to 71%) with the use of a scoring system based on PET metrics alone (PETVAS)[21]. In the current study, blinded radiology interpretation of PET/MRI (sensitivity 77%, specificity 88%) was better than the ability of PETVAS to correctly define disease activity. This suggests that the MRI component has value added benefit, specifically, the assessment of increased T2-weighted mural signal suggestive of vessel wall oedema. Importantly, we have shown a reduction in mural signal over time in patients achieving treatment-induced disease remission, suggesting that MRI metrics are modifiable. Thus, a scoring system that incorporates PET and MRI metrics might improve the accuracy of PET-only scores such as PETVAS.

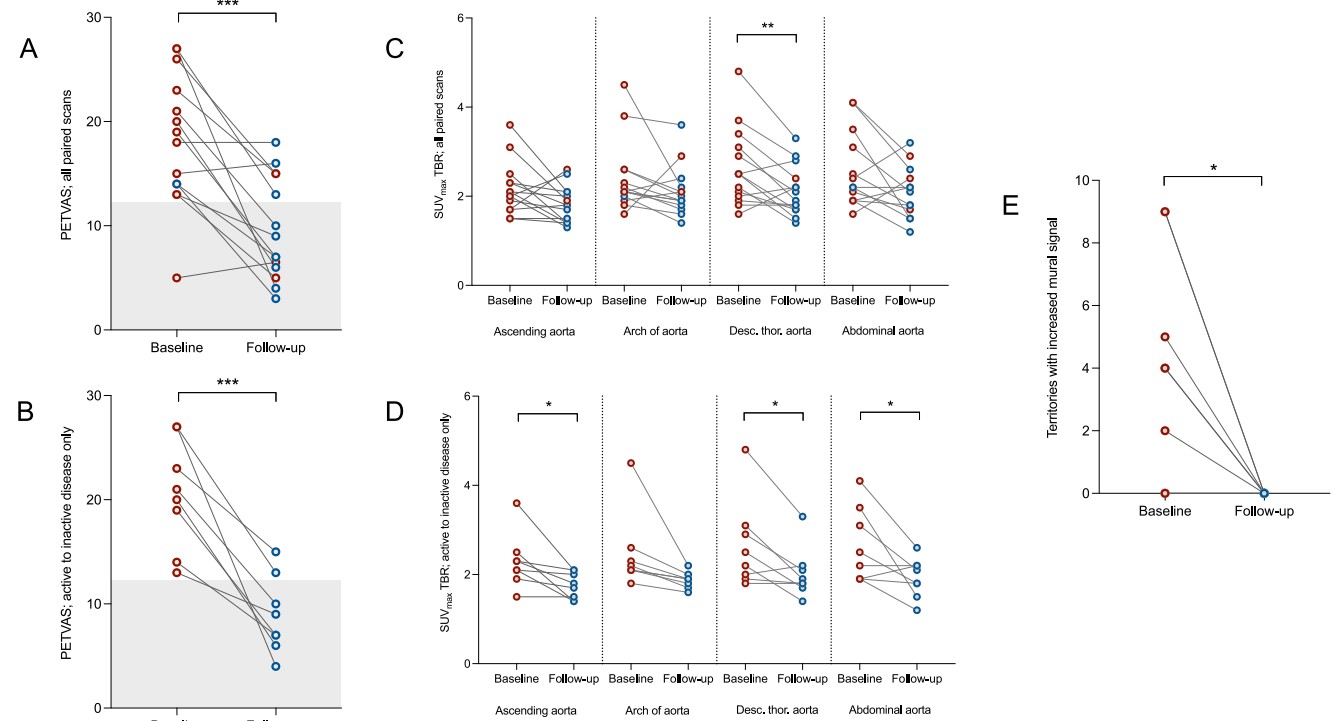

**Fig. 5 | PET/MRI metrics change longitudinally.** Changes in PETVAS (**A**, **B**), $SUV_{max}$ TBR (**C**, **D**) and T2-weighted mural signal (**E**) between baseline and final PET/MRI scans (**A**, $P = 0.0004$; **B**, $P = 0.0007$; **C** – descending thoracic aorta, $P = 0.008$; **D** – ascending aorta, $P = 0.02$; descending thoracic aorta, $P = 0.02$; abdominal aorta, $P = 0.04$; E, $P = 0.01$). Figures **A** and **C** include all participants who had >1 scan ($n = 14$). Figure **B**, **D** and **E** include only those with clinically active disease at baseline and inactive disease at follow-up ($n = 8$). The grey box indicates PETVAS cut-off for active versus inactive disease. Red dots represent patients with clinically active disease, blue dots represent clinically inactive disease. Analysis by two-sided paired $t$-test. Desc. thor. aorta descending thoracic aorta, PETVAS PET Vasculitis Activity Score, SUVmax maximum standardised uptake value, TBR target to background ratio.

Notably, several MRI metrics such as mural thickness and the presence of luminal abnormalities did not distinguish active from inactive disease in our patient cohort. Although the angiographic component of PET/MRI did not contribute to the assessment of disease activity here, it offers potential for monitoring vascular lesions (particularly in patients with TAK) over time and without radiation exposure, and so is an important component and significant advantage of this imaging modality. Several MRI-based 'vascular damage' scores are now available for use in LVV[22,23], and future work should consider combining disease activity with vascular damage indices to provide a more global assessment of an individual patient's disease.

Using clinical assessment of disease activity as the reference, the VAMP score combines PET metrics from 12 arterial territories with the presence or absence of T2-weighted mural signal in the aorta and other great vessels. Using a cut-off of ≥1 to indicate active disease, it was able to correctly categorise 90% of the scans in the derivation cohort as either active or inactive. Critically, the VAMP score is the first to utilise $SUV_{mean}$, which provides a more accurate representation of disease activity within a particular vascular segment and is more responsive to change in inflammatory activity over time compared to $SUV_{max}$. This is probably because of the potential for disease misclassification with $SUV_{max}$ due to patchy areas of high-grade FDG uptake, as seen with atherosclerosis[24]. Indeed, in the current study, several subjects with inactive GCA had high $SUV_{max}$ TBR values in the aorta and carotid arteries due to atheroma. The same subjects had lower $SUV_{mean}$ TBR values for the same territories. Conversely, some subjects with active disease demonstrated widespread yet relatively low-grade FDG uptake, often coupled with increased T2-weighted mural signal. Such disease would have been misclassified as inactive using $SUV_{max}$ but is classified correctly using $SUV_{mean}$, and likely represents smouldering vasculitic disease activity.

Most subjects with active disease identified through the presence of increased T2-weighted mural signal on MRI also had high-grade PET activity within the same territory, but not all. One subject with active disease would have been scored as inactive were it not for the additional score generated by the increased mural signal. Though PET metrics were a more powerful predictor of active disease, the important contribution that the MRI component makes to the VAMP score is underscored by logistic regression modelling which demonstrated an improved model fit when mural signal was added to PET metrics. Similar findings were reported by Einspieler et al. in a study of 12 patients with LVV[20]. Here, of 95 vascular segments identified as showing active vasculitis, 42% were identified by both PET and MRI, 48% were identified by PET alone, and 10% were identified by MRI alone. The authors also found that the correlation with CRP was stronger when combining PET and MRI metrics. While determination of T2-weighted mural signal forms an important part of our study, we recognise that its assessment remains more subjective than PET activity. Creating an objective method of quantifying mural oedema would be valuable, and artificial intelligence may have a role here.

Encouragingly, intra- and inter-operator concordance of VAMP scoring was high. This, alongside data showing excellent PET/MRI scan-rescan repeatability[25], provides reassurance that the score could be used reliably in both clinical and research settings. It should be noted that although we suggest a cut-off VAMP score of 1 to delineate active from inactive disease, this value is likely to be specific to this dataset and so should be viewed as a continuum. Accordingly, if a patient demonstrated a fall in VAMP score with treatment but this remained ≥1, this could still be viewed as a treatment success.

LVV comprises both GCA and TAK. Given the relative rarity of TAK, most studies of LVV incorporate both conditions together. Our cohort of 24 subjects included 14 with a diagnosis of GCA, six with TAK,

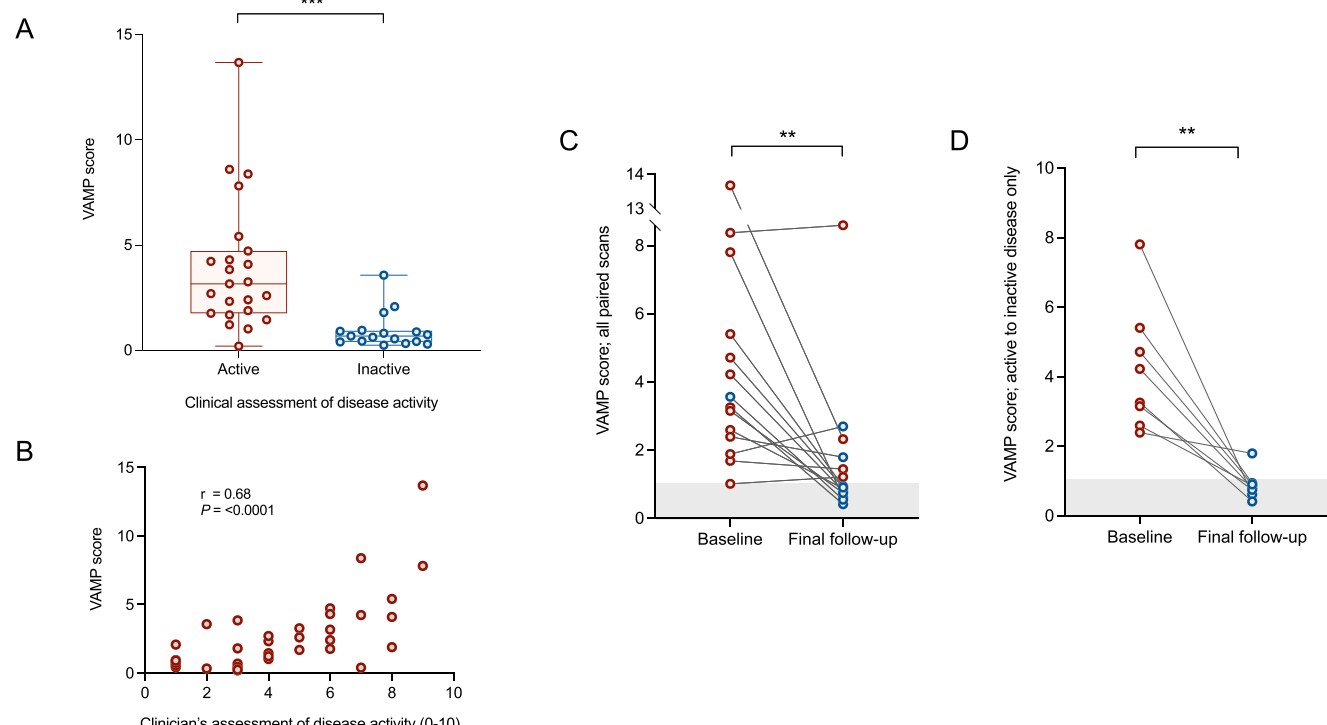

**Fig. 6 | Using the VAMP score to assess disease activity in LVV. A** VAMP score values in participants with active (n = 23) and inactive (n = 17) disease, as determined by clinical assessment (P ≤ 0.0001). **B** Correlation between VAMP score and clinical assessment of disease activity on a 0-10 scale (n = 39, P ≤ 0.0001). **C** Change in VAMP score from baseline scan to final follow-up scan for all participants who underwent more than one scan (n = 14, P = 0.007). **D** Change in VAMP score in those participants who were considered to have clinically active disease at baseline and inactive disease at follow-up (n = 8, P = 0.002). Red dots indicate clinically active disease.

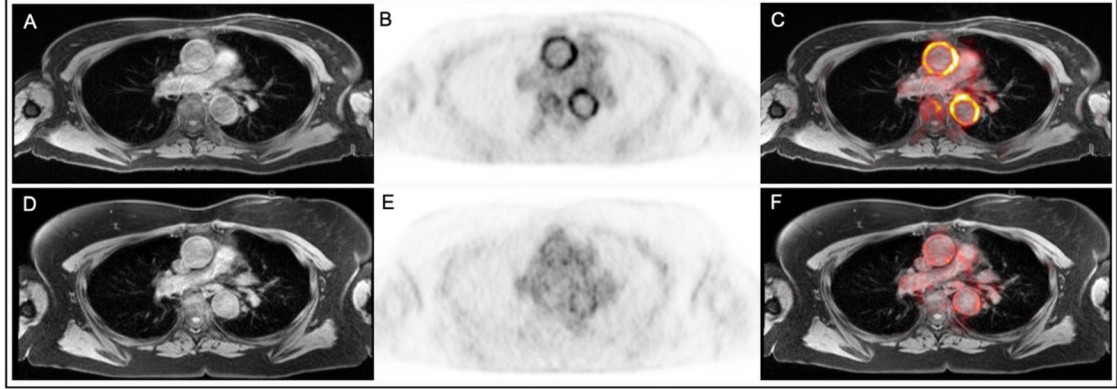

**Fig. 7 | Example of reduction in FDG uptake from baseline to follow-up.** Panels **A**–**C** are images obtained at baseline. Panels **D**–**F** are images obtained from the same participant at follow-up. Panels **A** and **D** show T1-weighted MR images, panels **B** and **E** show attenuation corrected PET, and panels **C** and **F** show fused PET/MR images.

## Table 2 | Correlations between PET/MRI scores and clinical metrics

|  | CRP (mg/dL) | ESR (mm/h) | Haemoglobin (g/L) | Platelets (x10⁹/L) | Graded clinical assessment | LRG1 (ng/mL) | Ang-2 (pg/mL) | Calprotectin (ng/mL) | Osteopontin (pg/mL) |
|---|---|---|---|---|---|---|---|---|---|
| VAMP score | 0.59*** | 0.52** | −0.44** | 0.42** | 0.68*** | 0.69*** | 0.80*** | 0.67*** | 0.64*** |
| PETVAS | 0.51*** | 0.51** | -0.42** | 0.27 | 0.65*** | 0.65*** | 0.76*** | 0.60*** | 0.61*** |

Values presented are r values denoting statistical correlation. Asterisks indicate statistical significance: * P < 0.05, **P < 0.01, ***P < 0.001. Analyses by two-sided Pearson's correlation coefficient (normally distributed data) or two-sided Spearman's correlation coefficient (non-normally distributed data). For VAMP score – CRP, P ≤ 0.0001; ESR P = 0.005; haemoglobin, P = 0.006; platelets, P = 0.009; graded clinical assessment, P ≤ 0.0001; LRG1, P ≤ 0.0001; Ang-2, P ≤ 0.0001; Calprotectin, P ≤ 0.0001; Osteopontin, P ≤ 0.0001. For PETVAS – CRP, P = 0.0008; ESR P = 0.006; hae-moglobin, P = 0.009; graded clinical assessment, P ≤ 0.0001; LRG1, P ≤ 0.0001; Ang-2, P ≤ 0.0001; Calprotectin, P = 0.0002; Osteopontin, P = 0.0002.
Ang-2 angiopoietin-2, CRP C-reactive protein, ESR erythrocyte sedimentation rate; LRG1 leucine-rich α-2 glycoprotein 1.

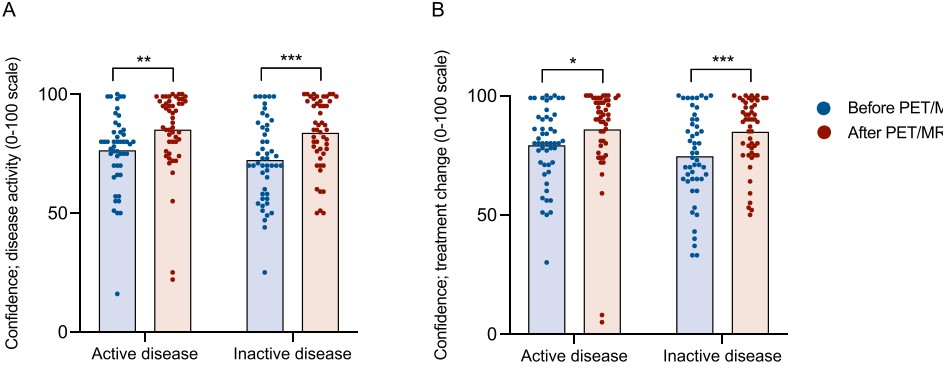

**Fig. 8 | PET/MRI improves physician confidence with managing LVV.** Confidence of participating clinicians (*n* = 10) in determination of disease activity (**A** – active disease, *P* = 0.008, *n* = 50 responses across five vignettes; inactive disease, *P* ≤ 0.0001, *n* = 50 responses across five vignettes) and decision to alter immuno-suppressive therapy (**B** – active disease, *P* = 0.03, *n* = 50 responses across five vignettes; inactive disease, *P* ≤ 0.0001, *n* = 50 responses across five vignettes) before and after disclosure of PET/MR imaging data across ten clinical vignettes (five vignettes depicting active disease and five depicting inactive disease). Data are presented as mean values + individual data points. Analysis by two-sided paired t-test.

and four without a firm diagnosis of one or the other. Given the differences in disease pathogenesis, clinical manifestations and treatments regimens between GCA and TAK[2], it would be surprising if disease biomarkers and scoring systems operated identically between the two. Indeed, our findings suggest that both PETVAS and VAMP are better suited to delineate disease activity in GCA than TAK. This is probably partly due to a higher stenosing potential in TAK, with patient symptoms (reflected in clinical assessments of disease activity) often attributable to chronic vascular lesions rather than active vascular inflammation. Indeed, clinical assessment of disease activity is more challenging in TAK. Also, inflammation is often confluent in GCA, affecting multiple arterial territories, whereas it is more often focal in TAK, again limiting the potential application of such a summary score in these patients. Future studies should focus on GCA and TAK separately with the latter requiring multi-centre, international collaboration. Such work could add modifications to the VAMP score depending on the underlying diagnosis, including the addition of the temporal arteries in GCA, or the renal and pulmonary arteries in TAK.

The temporal impact of disease treatment on any biomarker and scoring system is pertinent. Although we tried to scan subjects before treatment was initiated this was not always possible. At the time of the baseline PET/MRI scan, 50% of subjects included in our study were treatment-naïve; the other 50% had been receiving glucocorticoids for a median of four days. This reflects the real-world situation where clinicians start treatment empirically due to perceived risk of early disease complications without treatment. In our cohort, VAMP score was higher in those with active disease not on treatment versus those with active disease who had started treatment. Interestingly, however, we observed that both PETVAS and VAMP score remained abnormal in subjects with active disease treated with ≥1 week of high-dose glucocorticoids. In the same group, and as might be expected, CRP was <10 mg/dL in >80% of subjects. Contrary to previous reports[26], these findings suggest that PET, with or without MRI, may be less affected by treatment, at least in the early phases, which would improve its clinical utility even further, a finding which has been mirrored by other groups recently[27]. It should also be noted that, following baseline imaging, all treatment decisions were made by the referring team. Accordingly, participants received a variety of immunosuppressive treatments. Numbers were too small to discern any meaningful influence on imaging metrics of the different treatments received.

Although our study focussed on the ability of PET/MRI to monitor disease activity longitudinally in LVV, we performed exploratory analyses of six emerging serological biomarkers of disease; LRG1, Ang-2, soluble FMS-like tyrosine kinase 1 (sFlt1), calprotectin, osteopontin,

and endothelin-1 (ET-1). Plasma levels of LRG1, Ang-2, osteopontin and ET-1 were higher in active versus inactive LVV (LRG1: 68.3 ± 55.1 ng/ml vs. 37.7 ± 14.4 ng/ml, *P* = 0.04; Ang-2: 3,242 ± 1,648 pg/ml vs. 2,037 ± 536 pg/ml, *P* = 0.01; osteopontin: 109.1 ± 37.9 ng/ml vs. 65.8 ± 17.4 ng/ml, *P* = 0.0002; ET-1: 3.0 ± 1.3 pg/ml vs. 1.9 ± 1.0 pg/ml, *P* = 0.005). Plasma Ang-2 and calprotectin were higher in subjects with GCA compared to those with TAK (Ang-2: 3,111 ± 1,539 pg/ml vs. 1,787 ± 290 pg/ml, *P* = 0.02; calprotectin: 1,092 ± 636 ng/mL vs. 587 ± 297 ng/mL, *P* = 0.03). Interestingly, ET-1 was higher in those with TAK versus GCA (4.6 ± 1.0 pg/ml vs. 2.0 ± 0.7 pg/ml, *P* ≤ 0.0001). The study design did not allow us to compare these data to those from healthy subjects or disease controls, and this should be the focus of future work. We did observe strong correlations between plasma LRG1, Ang-2, calprotectin, and osteopontin and both PETVAS and VAMP score, perhaps providing some reassurance that PET/MRI was identifying truly active disease. Few studies have combined laboratory- and imaging-based biomarkers to assess disease activity in LVV. In 2021, Ma et al. combined PET/CT metrics with ESR and interleukin-2 receptor levels[28]. This approach was superior to established methods for identifying active TAK. This is an area of growing interest and future studies will determine the optimal method for fusing laboratory and imaging data for disease activity assessment in LVV.

We recognise some limitations to this work. This study utilised clinical assessment (history, examination, serological markers and previous imaging) as the 'gold-standard' reference for determination of disease activity, as has been adopted by previous studies in LVV[21,29]. Thus, the disease activity status of some subjects may have been incorrectly classified, particularly in those with TAK, and this is evident from a small number of PET/MRI scans that suggested active disease where clinical opinion suggested inactivity. Also, not all subjects had interval imaging. As this was a national study, this was mostly due to travel restrictions in place during the COVID-19 pandemic. Additionally, it should be noted that the VAMP score performed less well in the validation cohort that in the derivation cohort. Again, this is likely to relate in part to differences with clinical determination of disease activity, which was assessed using a 0–10-point scale in the validation cohort, rather than a binary (yes/no) assessment as in the derivation cohort. Furthermore, validation cohort images were obtained using a less advanced PET/MRI scanner and a different scanning protocol. As such, image quality was not directly comparable with the derivation cohort. Demographic differences and differences with treatment plans are also likely to exist, though are less likely to explain the discrepancies observed. Further validation of the VAMP score is required, and should be performed with the same scanner and imaging protocol to maximise reliability.

Our data suggest that PET/MRI has the potential to become a useful disease-monitoring tool in patients with LVV. Although access to PET/MRI remains limited, this may be an investigation to pursue in those patients where there is diagnostic uncertainty and through referral to experienced centres of excellence. As the spatial resolution of PET/MRI improves, it may be possible to include evaluation of the temporal arteries in an imaging protocol similar to the one used here[30–32]. Additionally, the development of novel PET radiotracers that target LVV-specific inflammation may allow greater confidence in discriminating LVV from other pathologies such as atherosclerosis[33]. Indeed, tracers with specific affinity for activated macrophages including [11]C-(R)-PK11195 have demonstrated promise in small studies[34]. Other potential candidates include ligands for translocator protein (TSPO)[35,36] and, more recently, the somatostatin receptor type 2 ligands [68]Ga-Dotatate and [18]F-FET-βAG-TOCA as part of an ongoing clinical study using PET/MRI[37,38].

## Methods
This national, prospective, observational cohort study was performed at the University of Edinburgh between July 2019 and February 2022. The study was conducted according to the principles of the Declaration of Helsinki and was approved by the Southeast Scotland Research Ethics Committee (Trial Registration: www.clinicaltrials.gov; NCT03914248).

### Study population
Adult subjects with a diagnosis of LVV (according to Chapel Hill Criteria[1]) were recruited across Scotland via the Scottish Systemic Vasculitis Network[39]. Gender was recorded on a self-reported basis. All subjects had suspected active (either new or relapsing) large vessel disease (i.e., not purely cranial GCA) at enrolment, and had received ≤3 weeks of high-dose glucocorticoids (≥20 mg prednisolone/day) immediately prior to starting the study. Subjects were excluded for any of the following reasons: (i) estimated glomerular filtration rate (eGFR) < 30 ml/min/1.73m$^2$; (ii) pregnant or breastfeeding; (iii) insulin-dependent diabetes mellitus; (iv) contraindication to PET or MRI; (v) unable to provide informed consent.

### Study design
**Study visits.** All subjects attended a baseline visit and were invited to return for a follow-up visit after 6 months. At each study visit, subjects attended the Edinburgh Imaging Facility where eligibility was confirmed, and written consent obtained. Subjects underwent clinical evaluation including assessment of height and weight. Following a 5-min rest period, seated systolic and diastolic blood pressure (BP) were recorded in duplicate from both arms using a validated oscillometric sphygmomanometer, the Omron HEM-705CP[40]. Clinical assessment of LVV disease activity was determined by the investigators prior to PET/MRI. Between study visits subjects returned to their referring clinical centre. All treatment decisions were made by the referring team, who were provided with detailed reports of the PET/MRI scans.

### Assessment of LVV disease activity
Given the lack of gold standard for determining LVV disease activity, clinical assessment of disease activity was used as the reference value for this study. Disease activity was assessed by the referring clinician, and independently by an investigating clinician (DP). Where there was disagreement, a third clinician would adjudicate (ND). Assessment was based on a combination of clinical history, physical examination, laboratory tests and any prior imaging. Disease activity was determined both as a binary 'yes/no', and also scored from 0 to 10 (0 = no perceived disease activity, 10 = maximal perceived disease activity).

### PET/MRI protocol
All subjects underwent hybrid [18]F-fluorodeoxyglucose (FDG) PET/MRI using a 3 T Biograph mMR PET/MRI scanner (Siemens Healthineers, Germany). Subjects were fasted for a minimum of 12 h prior to scanning. Plasma glucose was measured before FDG injection with <7 mmol/L considered acceptable. Sixty minutes prior to PET/MR imaging subjects were administered 400 MBq of intravenous FDG and then rested in a quiet, temperature-controlled room. Images were obtained during a 45-90-minute acquisition window over three bed positions, from top of head to thighs.

The imaging protocol was designed specifically for this study. Briefly, this involved a transverse black blood half-Fourier single-shot turbo spin echo (HASTE) sequence, an ECG-gated coronal T2 turbo spin echo (TSE) sequence, and a breath-held T1 VIBE sequence acquired in the transverse plane (both before and after gadolinium (Gadovist) contrast injection). Following completion of simultaneous PET and MRI acquisitions, a series of single-slice black blood 2D FLASH MR images were acquired with ECG-gating during a breath-hold to provide detailed delineation of the vessel wall at different sections of interest. Finally, a two-station MR angiogram was generated with coverage from the Circle of Willis to the femoral arteries. 0.2 mmol/kg of Gadobutrol (1.0 mmoL/ml) was administered in two injections triggered separately to ensure that vessels in the second angiogram were clearly visualised. Further details of the MR imaging protocol are given in Supplementary table 5.

### Overall assessment of PET/MRI
A radiologist experienced in interpreting both PET and MRI in vasculitis, and who was blinded to all clinical and demographic details, initially reported the scans as showing either active or inactive disease.

### Assessment of PET imaging
1. Visual comparison with liver uptake: First, the vascular tree was divided into 15 pre-determined arterial territories: ascending aorta, aortic arch, descending thoracic aorta, abdominal aorta, brachiocephalic artery and the right and left carotid, subclavian, vertebral, axillary, and iliac arteries. FDG uptake within each vascular territory was scored from 0 to 3 based on comparison with liver FDG uptake (0 = no uptake, 1 = uptake less than liver uptake, 2 = uptake equal to liver uptake, 3 = uptake greater than liver uptake).

2. PET Vasculitis Activity Score (PETVAS): Second, a cumulative PETVAS was calculated for each scan. PETVAS, first described by Grayson et al. combines the visual assessment scores from 9 of the 15 arterial territories (ascending aorta, aortic arch, descending thoracic aorta, abdominal aorta, brachiocephalic artery and right and left carotid and subclavian arteries) to provide a cumulative disease severity score from 0-27, with higher values indicating a greater burden of disease[21].

3. Maximum standardised uptake value (SUV$_{max}$) target-to-background ratio (TBR) scoring: Third, a SUV$_{max}$ TBR value was calculated for each of the 15 arterial territories. TBR scores are used to adjust for person-to-person variation in FDG avidity and background 'noise' by normalising arterial FDG uptake to a metabolically inactive reference value. Here, the point of maximal FDG uptake (SUV$_{max}$) within each arterial territory was identified by drawing region of interest (ROI) contours around sections of that segment. Next, a 'background' value was obtained from the venous bloodpool. This was performed by calculation of the mean SUV$_{mean}$ of three 5 mm radius spheres placed within the superior vena cava, inferior vena cava and right atrium. SUV$_{max}$ was then divided by the background SUV$_{mean}$ for each arterial territory to give individual TBR scores.

4. SUV$_{mean}$ TBR scoring: In addition to the established PET metrics described above, a novel LVV PET metric – SUV$_{mean}$ – was examined. SUV$_{mean}$ was determined by contouring the outer wall of each of 12 arterial territories (ascending aorta, aortic arch, descending thoracic aorta, abdominal aorta, and right and left carotid, subclavian, vertebral and axillary arteries) using FusionQuant software's centreline function (Supplementary figs. 10A & 10B). SUV$_{mean}$ was then calculated by dividing the total FDG uptake within a territory by the volume of the territory. For aortic territories, the inner wall was also contoured to capture FDG uptake within the lumen. This was then subtracted from the outer contour, thus excluding the lumen, and providing an SUV$_{mean}$ value for the wall only. SUV$_{mean}$ values for each arterial territory were then divided by a 'background' score (SUV$_{mean}$ venous bloodpool, as described above) to provide SUV$_{mean}$ TBR values. Finally, a total SUV$_{mean}$ TBR score was calculated by combining values from each of the 12 arterial territories.

All PET metrics including visual assessment of disease activity, PETVAS, SUV$_{max}$, SUV$_{mean}$, and background values were scored by a single operator blinded to the clinical details.

### Assessment of MR imaging

First, we assessed aortic mural thickness, with a thickness of ≥3 mm considered abnormal[16]. Second, the presence or absence of luminal abnormalities – including areas of stenosis, occlusion, dilation, or aneurysm – was noted. Finally, a qualitative assessment of increased mural signal on T2-weighted imaging and post-contrast mural enhancement on T1-weighted imaging was made (increased signal/enhancement either present or absent in each of the 12 arterial territories).

### Clinical correlations

PET and MRI metrics were correlated with clinical metrics including C-reactive protein (CRP), erythrocyte sedimentation rate (ESR), haemoglobin, platelet count, and graded clinical assessment of disease activity (on scale from 0-10).

### Emerging serological markers of LVV disease activity

Several potential serological markers of disease activity in LVV have shown promise. Leucine-rich α-2 glycoprotein 1 (LRG1) is upregulated in human models of neoangiogenesis, and proteomic analysis has suggested a role as a disease activity biomarker in Kawasaki disease, rheumatoid arthritis and small vessel vasculitis[41–44]. There are no data linking LRG1 to LVV. Angiopoietin-2 (Ang-2) is a growth factor which acts as part of the angiopoietin/Tie (tyrosine kinase with Ig and EGF homology domains) signalling pathway – a key regulator of angiogenesis. Previous studies have reported an upregulation of Ang-2 in GCA and associations between high circulating Ang-2 concentrations and an unfavourable disease course and high glucocorticoid requirements[45,46]. Soluble FMS-like tyrosine kinase 1 (sFlt1), a circulating form of vascular endothelial growth factor (VEGF)-receptor 1 which inhibits the angiogenic effects of VEGF and placental growth factor, is upregulated in placental vascular dysfunction and pre-eclampsia[47,48]. It has also been implicated in mouse models of complement activation and monocyte-driven inflammation[49,50], both important mechanistic drivers of LVV pathogenesis. Though upregulation of sFlt1 has been demonstrated in small vessel vasculitis[51], no study has examined sFlt1 in LVV. The S100A8/A9 heterodimer proteins, also known as calprotectin, are released locally by macrophages at sites of inflammation and as such may be of value in LVV. Previous work has demonstrated elevated plasma concentrations of calprotectin in GCA and its presence in GCA-affected arterial tissue[52]. Additionally, plasma levels do not appear to correlate with IL-6, suggesting potential efficacy in those treated with tocilizumab[46]. However, results in patients with TAK have been much less convincing, and further scrutiny is therefore required[52]. Osteopontin is a multifunctional glycoprotein expressed by a variety of innate and adaptive inflammatory cells and has been implicated in T helper (Th)1 and Th17 differentiation[53]. Mouse models of osteopontin overexpression demonstrate aortic medial thickening and neo-intimal formation[54]. Elevated serum osteopontin levels have been demonstrated in active GCA[55], however, no study has evaluated osteopontin as a disease biomarker in TAK. Endothelin-1 (ET-1) is a potent endogenous vasoconstrictor produced by the endothelium[56]. Through action on both endothelin-A and endothelin-B receptors ET-1 acts reciprocally with nitric oxide (NO) to regulate systemic vascular function[57]. By damaging the endothelium, LVV may upset NO/ET-1 balance promoting arterial stiffening and endothelial dysfunction. ET-1 upregulation has been demonstrated in temporal artery tissue from patients with GCA[58], however, the role of the endothelin system in LVV remains largely unknown.

LRG1, Ang-2, sFlt1, calprotectin, osteopontin and ET-1 were measured in plasma using ELISA. For LRG1 (IBL-America, Minneapolis, USA): mean recovery of LRG1 from plasma: 107%; assay cross-reactivity <0.01% with human Aβ (1-42), human sAPPβ and human sAPPα; intra- and inter-assay coefficients of variation: 4%. For Ang-2: (R&D Systems, Minneapolis, USA): mean recovery of Ang-2 from plasma: 95%; no significant cross-reactivity; intra- and inter-assay coefficients of variation: 6% and 9%, respectively. For sFlt1: (R&D Systems, Minneapolis, USA): mean recovery of sFlt1 from plasma: 99%; no significant cross-reactivity; intra- and inter-assay coefficients of variation: 3% and 7%, respectively. For calprotectin: (R&D Systems, Minneapolis, USA): mean recovery of calprotectin from plasma was not reported, but was 100% from urine and 85% from cell culture media; no significant cross-reactivity; intra- and inter-assay coefficients of variation: 3% and 5%, respectively. For osteopontin: (R&D Systems, Minneapolis, USA): mean recovery of osteopontin from plasma: 102%; no significant cross-reactivity; intra- and inter-assay coefficients of variation: 3% and 6%, respectively. For ET-1: (R&D Systems, Minneapolis, USA): mean recovery of ET-1 from plasma: 93%; no significant cross-reactivity; intra- and inter-assay coefficients of variation: 3% and 6%, respectively.

### Development of a novel scoring system – the Vasculitis Activity using MR PET (VAMP) score

Logistic regression modelling was used to determine which PET and MRI metrics were most strongly associated with the presence of clinically active disease. Metrics were first assessed independently using univariable logistic regression analysis. Based on the findings of the univariable analysis, multivariable logistic regression models were constructed to determine how metrics of interest performed in combination to predict active disease. Initially, the single most highly associated PET and MRI metrics were selected. Additional metrics with $P$ values < 0.05 from univariable analysis were then added to the model sequentially to determine if this improved model fit or led to overfit. The relative performance of each model (goodness-of-fit) was determined by reviewing Akaike's Corrected Information Criterion (AICc) and the Variance Inflation Factor (VIF). The results of both univariable and multivariable models are presented as odds ratios (ORs) with 95% confidence intervals (CI). Where relevant, continuous variables were transformed before inclusion in the analysis.

The relative importance of each selected metric was calculated from the logistic regression models and a weighting applied based on ORs. This weighting was then used to create a new scoring system, the **V**asculitis **A**ctivity using **M**R **P**ET (VAMP) score. Simplification of the weighting system was performed to enable calculation of the VAMP score without the need for an automated calculator with the purpose of improving clinical utility, as has been demonstrated elsewhere in the literature[22]. Receiver operating characteristic (ROC) curve analysis was used to characterise the ability of the VAMP score to distinguish active from inactive disease. A correlation between the VAMP score and

numerical disease activity assessment (0 = no perceived disease activity, 10 = maximal perceived disease activity) was also performed. Scans were scored independently by two trained operators. A subset of 10 scans, selected at random, were also re-scored by the same operator after ≥6 weeks (to reduce recall bias). Intra- and inter-operator reliability were assessed using Pearson's correlation coefficient and Bland-Altman analysis.

## Validation of the VAMP score

The VAMP score was applied to an independent cohort of patients with LVV attending the National Institutes of Health (NIH) in Bethesda, Maryland. Sixty-four scans in 27 subjects were obtained at different timepoints throughout the clinical course. The PET/MRI protocol used at the NIH been described in detail previously[40]. Clinical assessment of disease activity was performed prior to each scan using a physician's global assessment (PGA) scale from 0-10 (0 = no perceived disease activity, 10 = maximum perceived disease activity). All scans were analysed by a single operator (DP) blinded to clinical data and previous PETVAS. VAMP score and PETVAS were calculated, and values were then compared with PGA.

## Assessment of clinical utility of PET/MRI in LVV

Finally, we determined if the imaging data generated by PET/MRI would support clinicians' management of patients with LVV. Ten case vignettes (five GCA, five TAK) were designed based on a selection of patients previously reviewed and imaged at our centre. Clinical details including history, examination findings, blood results and imaging studies were summarised and presented in each vignette (available in Supplementary Methods). Vignettes were distributed via an online survey platform (SurveyMonkey, Momentive inc., USA) to an international panel of clinicians experienced in the management of patients with LVV. For each vignette, participating clinicians were asked to evaluate disease activity and whether management should be altered, in addition to rating confidence in both decisions. Clinicians were then provided the findings of a PET/MRI scan, though without a radiologist's summative opinion regarding disease activity. Following disclosure of PET/MRI metrics, clinicians were then asked the same questions for a second time. Responses were collated and analysed using descriptive analyses.

## Imaging and statistical analysis

Imaging data were analysed using Syngo.via (Siemens, Germany) and FusionQuant (Cedars-Sinai Medical Centre, USA) software. Statistical analyses were performed using Prism (GraphPad Software Inc, USA). For cross-sectional analyses, two-way ANOVA with Šidák's multiple comparison test was used to compare metrics across multiple arterial territories (visual comparison with liver, $SUV_{max}$, and $SUV_{max}$ TBR). Unpaired $t$-tests were used to compare cumulative metrics (PETVAS and MRI metrics). For paired analyses, either paired $t$-tests or Mann–Whitney tests were used depending on normality of the data. Pearson's or Spearman's correlation coefficients were used as appropriate for assessment of correlation between imaging metrics and clinical metrics. Significance was set at $P < 0.05$.

## Reporting summary

Further information on research design is available in the Nature Portfolio Reporting Summary linked to this article.

## Data availability

Source data are provided with this paper. De-identified individual participant data are available from the corresponding author (bean.dhaun@ed.ac.uk) from the publication date. Source data are provided with this paper.

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

## Acknowledgements

D.Pugh is funded by a Clinical Academic Fellowship from the Chief Scientist Office, Scotland (CAF/19/01). N.D. is supported by a Senior Clinical Research Fellowship from the Chief Scientist Office (SCAF/19/02).

## Author contributions

N.D., N.B., D.Pugh, G.M. and D.Patel designed the study; D.Pugh, D.Patel and J.D. performed PET/MRI image analysis; D.Pugh, A.C. and L.B. performed laboratory analysis; P.C.G., M.A. and S.T. provided the validation cohort. P.J.G. provided additional statistical analysis. N.D., N.B. and D.Pugh wrote the manuscript; all authors provided critical appraisal and approval for the final manuscript.

## Competing interests

The authors declare no competing interests.
