## [Peer Review File · Nature Communications]

18F-FDG-PET/MR imaging to monitor disease activity in large vessel vasculitisREVIEWER COMMENTS

Reviewer #1 (Remarks to the Author):

The authors describe a prospective, longitudinal clinical study investigating the use of hybrid FDG-PET and MRI/MRA in the assessment of patients with large vessel vasculitis (LVV).

Key Findings:

1. Blinded radiology assessment of PET/MRI could distinguish active from inactive LVV with 78% sensitivity and 88% specificity, compared to compared to clinical assessment, the current (and I would suggest, inadequate) 'gold standard'.
2. Both PET-derived (PETVAS, SUVmean>max TBR) and MR-derived (T2-weighted mural signal) metrics contributed to this assessment.
3. These metrics changed over time and with treatment, as patients progressed from active to inactive disease.
4. This led to development of the VAMP score (with high inter-operator reliability); ROC analysis demonstrated an AUC of 0.90 ($P < 0.0001$) with a suggested cut-off of 1 - sensitivity 96%, specificity 82% – to distinguish active from inactive disease; VAMP performed better than PETVAS; though less well in an external validation cohort.
5. PET/MRI improved clinician confidence in assessing disease activity and altering treatment.

The authors are to be commended for performing this study in a rare group of diseases (especially noting that recruitment was conducted during the COVID-19 pandemic). These diseases are difficult to treat, have no diagnostic criteria (only classification criteria) and no reliable measures of disease activity. In this context, there have been increasing recommendations within global guidelines for research into the use of imaging modalities. The issues to date have centred around imaging that is both accurate – in terms of disease activity assessment – and which utilises minimal/optimal radiation dose given these diseases are lifelong, often affecting young people, and so the risk of excessive radiation exposure is real.

There are several novel aspects to the study. First, this is one of the first longitudinal studies that assesses the PET/MRI using a clinically used tracer, ^{18}F -FDG, and the authors developed

a bespoke PET/MRI imaging protocol that has not previously described for these diseases. (The few studies in LVV that have assessed PET/MRI, e.g. a study earlier this year examining the use of a novel somatostatin analogue, have utilised assessments designed for PET/CT).

Comments/Questions

Please provide a table comparing patients with active disease at first imaging versus inactive disease at follow-up imaging (not just baseline and follow-up comparisons as in Table 1); this is needed to understand how well matched the active/inactive groups are in subsequent analyses.

Are there additional clinical phenotype data, not reported in Table 1, that could be important in interpreting the changes in interval imaging (e.g. presence of established CV disease, diabetes, other CV risk factors).

Can the authors explain the reasons why the VAMP score operated less well in the validation cohort, particularly considering the high inter-operator reliability they have reported? This suggests potential differences in the patient cohorts, their treatments, timing of scans, or technical imaging parameters – do they have any of these data to explore this in more detail?

The patients in their study received a variety of immunosuppressive treatments; whilst numbers are small, was there any suggestion of differential imaging response to these treatments (e.g. tocilizumab versus conventional DMARDS)?

The exploratory biomarker data are intriguing. Why were these two biomarkers chosen for the study (of the many putative biomarkers reported in the literature, e.g. IL6, pentraxin 3, the S100 proteins, osteopontin, YKL40). Did they measure any of these in their cohort?

Given their expertise, could the authors explain whether they think that PET/MRI might be useful in all patients with LVV or specific subgroups with either GCA with large vessel involvement or those with TAK – as some parameters in their study seemed to have better

discrimination in GCA versus TAK?

Reviewer #2 (Remarks to the Author):

This small observational study involved 14 patients who underwent PET-MRI scans both at baseline and six-month follow-up. Patients had giant cell arteritis (GCA), Takayasu arteritis (TAK), or undifferentiated large-vessel vasculitis, including newly diagnosed cases and those experiencing relapses.

1. You are proposing a novel score for PET/MRI. Please describe this score in the abstract, specifying also the relevant arteries.
2. Please provide separate results for GCA and TAK.
3. Please provide separate results for new-onset and relapsing disease, acknowledging the limited sample size for drawing conclusions within subgroups.
4. Elaborate on the characteristics of the four patients with undifferentiated large-vessel vasculitis, and explain why you could not classify them as GCA or TAK. Consider the possibility of excluding these patients from calculations.
5. Given the ability of MRI and PET to detect vasculitis in smaller cranial arteries, such as temporal and occipital arteries, consider incorporating them into a scoring system of GCA.
6. Your results may have been biased by including patients on treatment at baseline.
7. The EULAR recommendations on imaging in large-vessel vasculitis have been updated in 2023.

REVIEWER 1

The authors describe a prospective, longitudinal clinical study investigating the use of hybrid FDG-PET and MRI/MRA in the assessment of patients with large vessel vasculitis (LVV).

Key Findings:

1. Blinded radiology assessment of PET/MRI could distinguish active from inactive LVV with 78% sensitivity and 88% specificity, compared to clinical assessment, the current (and I would suggest, inadequate) 'gold standard'.
2. Both PET-derived (PETVAS, SUV_{mean}>max TBR) and MR-derived (T2-weighted mural signal) metrics contributed to this assessment.
3. These metrics changed over time and with treatment, as patients progressed from active to inactive disease.
4. This led to development of the VAMP score (with high inter-operator reliability); ROC analysis demonstrated an AUC of 0.90 ($P<0.0001$) with a suggested cut-off of 1 – sensitivity 96%, specificity 82% – to distinguish active from inactive disease; VAMP performed better than PETVAS; though less well in an external validation cohort.
5. PET/MRI improved clinician confidence in assessing disease activity and altering treatment.

The authors are to be commended for performing this study in a rare group of diseases (especially noting that recruitment was conducted during the COVID-19 pandemic). These diseases are difficult to treat, have no diagnostic criteria (only classification criteria) and no reliable measures of disease activity. In this context, there have been increasing recommendations within global guidelines for research into the use of imaging modalities. The issues to date have centred around imaging that is both accurate – in terms of disease activity assessment – and which utilises minimal/optimal radiation dose given these diseases are lifelong, often affecting young people, and so the risk of excessive radiation exposure is real.

We thank the Reviewer for recognising the importance of our study in a rare disease which has no diagnostic criteria, and where recent guideline-based recommendations have urged research into the use of multi-modality imaging to assess disease activity. We also appreciate the Reviewer's recognition in the difficulties of undertaking a study like this during the COVID pandemic.

There are several novel aspects to the study. First, this is one of the first longitudinal studies that assesses the PET/MRI using a clinically used tracer, 18-FDG, and the authors developed a bespoke PET/MRI imaging protocol that has not previously described for these diseases. (The few studies in LVV that have assessed PET/MRI, e.g. a study earlier this year examining the use of a novel somatostatin analogue, have utilised assessments designed for PET/CT).

Thank you for these positive comments. We value the Reviewer's recognition that our study is novel not only because of its longitudinal nature and the development of a bespoke PET/MRI protocol, but also because of its clinical relevance given the use of the widely available radiotracer (¹⁸F-FDG).

1. Please provide a table comparing patients with active disease at first imaging *versus* inactive disease at follow-up imaging (not just baseline and follow-up comparisons as in Table 1); this is needed to understand how well matched the active/inactive groups are in subsequent analyses.

We thank the Reviewer for this helpful suggestion. We have added this table to the manuscript (Table 1 in the revised manuscript), and it is also presented below (Table i).

Table i. Subject characteristics stratified by clinical assessment of disease activity.

Data are presented as n (%), mean±SD, or median [IQR]. *Ang-2*, *angiopoietin-2*; *BP*, *blood pressure*; *ESR*, *erythrocyte sedimentation rate*; *GCA*, *giant cell arteritis*; *LRG1*, *leucine-rich α -2 glycoprotein 1*; *MMF*, *mycophenolate mofetil*; *TAK*, *Takayasu arteritis*.

Parameter	Baseline			Follow-up		
	Active disease based on clinician assessment	Inactive disease based on clinician assessment	Total	Active disease based on clinician assessment	Inactive disease based on clinician assessment	Total
n	17	7	24	6	10	16
Age, years	57±15	70±10	61±15	54±21	63±12	60±16
Female (%)	13 (76)	4 (57)	17 (71)	5 (83)	7 (70)	12 (75)
White (%)	17 (100)	7 (100)	24 (100)	6 (100)	10 (100)	16 (100)
Smoking (%)						
	Current	4 (24)	0 (0)	4 (17)	1 (10)	2 (13)
	Ex-	3 (18)	4 (57)	7 (29)	2 (20)	2 (13)
	Never	10 (59)	3 (43)	13 (54)	7 (70)	12 (75)
Diabetes (%)	0 (0)	0 (0)	0 (0)	0 (0)	0 (0)	0 (0)
Pre-existing hypertension (%)	2 (12)	5 (71)	7 (29)	0 (0)	2 (20)	2 (13)
Disease Characteristics						
Diagnosis (%)						
	GCA	13 (76)	1 (14)	14 (58)	3 (50)	9 (90)
	TAK	4 (24)	2 (29)	6 (25)	3 (50)	1 (10)
	Unspecified	0 (0)	4 (57)	4 (17)	0 (0)	0
Disease duration, days	100 [39-286]	88 [9-986]	94 [38-360]	444 [314-711]	397 [225-572]	418 [252-543]
Immunosuppression at time of PET/MRI (%)						
	Prednisone	9 (53)	3 (43)	12 (50)	4 (67)	10 (100)
	Methotrexate	1 (6)	1 (14)	3 (13)	2 (33)	2 (20)
	MMF	1 (6)	0 (0)	1 (4)	1 (17)	1 (10)
	Tocilizumab	0 (0)	0 (0)	0	4 (67)	2 (20)
	Nil	7 (41)	4 (57)	11 (46)	0 (0)	0
Duration of immunosuppression, days	5 [0-18]	0 [0-420]	4 [0-19]	190 [163-383]	213 [180-390]	205 [180-385]
Clinical						
Systolic BP, mmHg	137±19	140±21	138±19	154±33	145±23	148±27
Diastolic BP, mmHg	80±14	75±9	79±13	85±19	80±16	82±17

Heart rate, bpm	83±11	86±21	84±14	78±14	73±15	75±14
Body mass index, kg/m ²	23.9±3.9	28.3±4.7	25.2±4.5	24.1±2.4	25.2±3.5	25.1±3.0
Laboratory						
Hemoglobin, g/L	123±15	128±29	125±18	133±16	141±15	138±15
Leucocytes, x10 ⁹ /L	10.9±3.5	10.4±3.6	10.7±3.5	10.1±5.1	8.8±2.5	9.3±3.5
Platelets, x10 ⁹ /L	417±133	416±223	417±159	320±60	307±116	311±98
C-reactive protein, mg/L	18 [2-66]	11 [3-69]	13 [3-63]	1 [0-7]	2 [1-12]	2 [0-10]
ESR, mm/h	53±36	43±41	49±37	13±9	10±6	11±7
Creatinine, mg/dL	0.70±0.12	0.93±0.20	0.77±0.18	0.79±0.16	0.75±0.06	0.78±0.12
LRG1, ng/mL	72.2±57.7	61.8±28.3	69.5±51.3	39.2±18.6	35.7±14.3	36.8±15.2
Ang-2, pg/mL	3,443±1,714	3,107±1,512	3,365±1,646	2,503±896	1,996±597	2,154±715

2. Are there additional clinical phenotype data, not reported in Table 1, that could be important in interpreting the changes in interval imaging (e.g., presence of established CV disease, diabetes, other CV risk factors).

Thank you for raising this important point. In response to the Reviewer's suggestions, we have now collected data relating to the presence of pre-existing hypertension, diabetes mellitus, and previous cardiovascular events for all subjects recruited to our studies. We have added the presence of pre-existing hypertension and diabetes to **Table i** (see above). Interestingly, none of the participants included in the study had a pre-existing diagnosis of diabetes mellitus, though this will be influenced by the fact that the protocol excluded those with insulin-dependent diabetes (though not those with diet-controlled diabetes mellitus or those on oral agents). With regards to pre-existing cardiovascular events, it is difficult to determine whether such events were related to cardiovascular disease which pre-dated the diagnosis of large vessel vasculitis (LVV), or indeed whether such events (e.g., stroke) were a consequence of LVV. Given this doubt, we have elected not to include these data within the manuscript.

We have also performed new analyses which suggest that neither the magnitude of blood pressure (systolic or diastolic, systolic presented below in **Figure i**), nor a pre-existing diagnosis of hypertension (**Figure ii**) appeared to influence imaging metrics, using the VAMP score as a summary of this.

Figure i. No significant correlation observed between systolic BP and disease activity as measured by VAMP score. Analysis by Pearson's correlation coefficient.

Figure ii. No significant difference observed in disease activity based on a diagnosis of pre-existing hypertension. Analysis based on baseline imaging only. Analysis by Mann-Whitney test.

3. Can the authors explain the reasons why the VAMP score operated less well in the validation cohort, particularly considering the high inter-operator reliability they have reported? This suggests potential differences in the patient cohorts, their treatments, timing of scans, or technical imaging parameters – do they have any of these data to explore this in more detail?

Thank you for raising this important point. When undertaking this study, we considered it crucial to validate our findings in an external cohort. However, currently, there are few research teams examining the use of positron emission tomography with magnetic resonance imaging (PET/MRI) in patients with LVV, which itself highlights the novelty of our work. Given both our imaging protocol and prospective study design were novel, we found it challenging to find a suitable comparator group. Through collaboration with Dr Peter Grayson at the National Institutes of Health (NIH), we were able to use the validation cohort presented in our manuscript. However, we recognise its limitations which include differences in the scanning protocol to our own derivation cohort as well as in the approach to the clinical assessment of disease activity. Specifically, the validation cohort were scanned using a less advanced PET/MRI scanner and a different scanning protocol. As such, image quality was not directly comparable to our own scans. As PET/MRI scanners improve and scanning protocols advance, these problems should become less frequent. With regards to disease activity, this was assessed clinically on a 0–10-point scale, rather than a binary (yes/no) assessment as in the derivation cohort. This makes direct comparisons difficult. We have tried to be clear in the manuscript that clinical assessment of disease activity is far from ideal, and we feel that this is reflected more in the validation cohort than the derivation cohort. Demographic differences and differences with treatment plans are also likely to exist, though are less likely to explain the discrepancies observed. We agree that further validation of the VAMP score is required, though this should ideally be performed with the same scanner and imaging protocol to maximise reliability. We have updated the Discussion section of the manuscript to include the details above.

4. The patients in their study received a variety of immunosuppressive treatments; whilst numbers are small, was there any suggestion of differential imaging response to these treatments (e.g., tocilizumab versus conventional DMARDS)?

This is an important comment which we apologise for not addressing in the original manuscript. As the Reviewer suggests, the numbers here are too small to draw meaningful conclusions. However, we present these data below:

Table ii. Immunosuppressive agents initiated following baseline study visit.

Treatment initiated following baseline scan	n	Clinically active disease at follow-up	Clinically inactive disease at follow-up
Prednisone	14	4 (29)	10 (71)
Methotrexate	4	2 (50)	2 (50)
MMF	2	1 (50)	1 (50)
Tocilizumab	6	4 (67)	2 (33)
Nil	0	0 (0)	0 (0)

While not statistically significant, any trends observed here may be influenced by those with a greater burden of disease at time of baseline imaging having a higher likelihood of being treated more aggressively from the outset. We have updated the Discussion section of the revised manuscript to reflect these findings, while acknowledging that numbers are too small for conclusions to be drawn.

5. The exploratory biomarker data are intriguing. Why were these two biomarkers chosen for the study (of the many putative biomarkers reported in the literature, e.g., IL6, pentraxin 3, the S100 proteins, osteopontin, YKL40). Did they measure any of these in their cohort?

We thank the Reviewer for raising this. In the original manuscript we include data supporting the ability of both angiopoietin-2 (Ang-2) and leucine-rich glycoprotein-1 (LRG1) to distinguish active from inactive LVV and show strong correlations of these biomarkers with PET/MRI metrics. Our choice of biomarkers examined was based on review of the literature together with consideration of the likely mechanisms of inflammation initiation and propagation in LVV. Thus, in addition to Ang-2 and LRG1, we analysed soluble FMS-like tyrosine kinase 1 (sFlt-1), the S100A8/A9 heterodimer proteins (calprotectin), osteopontin, and endothelin-1 (ET-1).

sFlt-1, a circulating form of vascular endothelial growth factor (VEGF)-receptor 1 which inhibits the angiogenic effects of VEGF and placental growth factor, is upregulated in placental vascular dysfunction and pre-eclampsia.^{1, 2} It has also been implicated in mouse models of complement activation and monocyte-driven inflammation,^{3, 4} both important mechanistic drivers of LVV pathogenesis. Though upregulation of sFlt1 has been demonstrated in small vessel vasculitis,⁵ no study has examined sFlt1 in LVV.

Calprotectin is released locally by macrophages at sites of inflammation and as such may be of value in LVV. Previous work has demonstrated elevated plasma concentrations of calprotectin in GCA and its presence in GCA-affected arterial tissue.⁶ Additionally, plasma levels do not appear to correlate with IL-6, suggesting potential efficacy in those treated with tocilizumab.⁷ However, results in patients with TAK have been much less convincing, and further scrutiny is therefore required.⁶

Osteopontin is a multifunctional glycoprotein expressed by a variety of innate and adaptive inflammatory cells and has been implicated in Th1 and Th17 differentiation.⁸ Mouse models of osteopontin overexpression demonstrate aortic medial thickening and neo-intimal formation.⁹ Elevated serum osteopontin levels have been demonstrated in active GCA,¹⁰ however, no study has evaluated osteopontin as a disease biomarker in TAK.

ET-1 is a potent endogenous vasoconstrictor produced by the endothelium.¹¹ Through action on both endothelin-A and endothelin-B receptors ET-1 acts reciprocally with nitric oxide (NO) to regulate systemic vascular function.¹² By damaging the endothelium, LVV may upset NO/ET-1 balance promoting arterial stiffening and endothelial dysfunction. ET-1 upregulation has been demonstrated in temporal artery tissue from patients with GCA¹³, however, the role of the endothelin system in LVV remains largely unknown.

Plasma concentrations of osteopontin and ET-1 were higher in active LVV compared with inactive LVV (osteopontin: 109.1 ± 37.9 ng/ml *versus* 65.8 ± 17.4 ng/ml, $P=0.0002$; ET-1: 3.0 ± 1.3 pg/ml *versus* 1.9 ± 1.0 pg/ml, $P=0.005$) (**Figure iii**), with no differences observed in sFlt-1 or calprotectin. We also found strong correlations between calprotectin and osteopontin and PET/MRI metrics (**Figure iv**). These data are presented below:

Figure iii. Plasma osteopontin and endothelin-1 were higher in active disease *versus* inactive disease. Analyses by unpaired t test.

Figure iv. Correlations between plasma calprotectin and plasma osteopontin and the PET/MRI-derived VAMP score. Analyses by Pearson's correlation coefficient.

We have now updated the manuscript to include these new data (**Supplementary figure 5**) and discussion around these and their future promise.

6. Given their expertise, could the authors explain whether they think that PET/MRI might be useful in all patients with LVV or specific subgroups with either GCA with large vessel involvement or those with TAK – as some parameters in their study seemed to have better discrimination in GCA versus TAK?

We thank the Reviewer for this important comment, and we agree that PET/MRI is likely to be more beneficial in certain sub-populations of LVV; as an author group, we have given this a great deal of thought. PET/MRI offers a unique opportunity to obtain highly detailed anatomical data together with a multi-modality assessment of inflammatory activity. Patients with both GCA and TAK benefit differently from the various components of the hybrid scan. For example, in those patients with large vessel GCA, the PET component is arguably more important, as widespread, high-grade FDG uptake might be expected, and changes in disease activity might be best monitored using this aspect of the scan (**Figure v**).

Figure 5. Changes in aortic and subclavian artery FDG uptake over time in a patient with LV-GCA.

However, in our opinion, it might be insufficient to follow up some of the patients with large vessel GCA included in this study with PET alone, as several had important areas of vascular stenosis which were responsible for symptoms, and which were difficult to differentiate from active disease (**Figure vi**).

Figure 6. Area of large vessel stenosis in patients with LV-GCA. This lesion was responsible for symptoms though may have been missed with PET alone as it was not acutely inflamed.

In a patient with TAK, it may be that the MR angiography component of the scan is most valuable, in order to track vessel damage over time, as TAK has a greater stenosing potential than GCA. Despite this, the same patient with TAK may have discrete areas of FDG uptake demonstrating important disease activity necessitating changes in treatment, which might be missed with MRA alone. We present an example of this below (**Figure vii**).

Figure vii. Panel A demonstrates high grade FDG uptake affecting the origin of the superior mesenteric artery (arrow) in a patient with TAK. Panel B shows resolution of FDG uptake at follow-up.

As such, it is likely that PET/MRI has value-added-benefit in both GCA and TAK, for different reasons, and, in the future, may have a place in the longitudinal disease monitoring in most patients with LVV. Given our novel VAMP score is largely assessing disease activity and not discriminating symptom burden from established damage, it may be more applicable in its current form to those with GCA rather than TAK. In future, it may be useful to add modifications to the score depending on whether the patient has GCA or TAK (e.g., adding data from renal or pulmonary arteries to those with TAK, or from temporal arteries to those with GCA).

REVIEWER 2

This small observational study involved 14 patients who underwent PET-MRI scans both at baseline and six-month follow-up. Patients had giant cell arteritis (GCA), Takayasu arteritis (TAK), or undifferentiated large-vessel vasculitis, including newly diagnosed cases and those experiencing relapses.

We thank the Reviewer for their valuable insights into our work. We have responded to each of their comments individually below. However, respectfully, we would like to point out that our study included 24 patients and not 14 patients; these subjects underwent a total of 40 PET/MRI scans. As recognised by Reviewer 1, large vessel vasculitis (LVV) is a rare disease, and prospective clinical studies will always be limited by sample size. Indeed, in terms of sample size, our study compares favourably with others in the field (**Table iii**). Despite its modest size, our study was able to answer its original question and sets the scene for (ideally) larger and longer studies in the field.

Table iii. Comparative study size of previous studies in the field.

Study	Study size
Clemente, Rheumatology 2022 ¹⁴	17 patients with TAK
Laurent, Sci Rep 2019 ¹⁵	13 patients with LVV
Reichenbach, Rheumatology 2018 ¹⁶	30 patients with GCA
Einspieler, Eur J Nucl Med Mol Imaging 2015 ¹⁷	12 patients with LVV
Stellingwerff, Medicine 2015 ¹⁸	18 patients with GCA + control groups

1. You are proposing a novel score for PET/MRI. Please describe this score in the abstract, also specifying the relevant arteries.

Thank you for this comment. We were previously limited by the Journal's suggested word limit for the abstract. However, at the Reviewer's suggestion we have modified this. The updated methods section of the abstract now reads as follows:

Methods

Patients with active LVV were recruited to a prospective, observational study and underwent simultaneous ¹⁸F-fluorodeoxyglucose (FDG)-PET and gadolinium-enhanced MRI at baseline and ≥6 months alongside clinical phenotyping. PET/MRI scans were assessed for disease activity and referenced to clinical assessment. Using logistic-regression modelling of PET/MRI metrics, we developed a novel PET/MRI-specific Vasculitis Activity using MR PET (VAMP) score, which we evaluated in an independent validation cohort. The VAMP score utilised PET and MRI metrics from arterial territories of interest (ascending aorta, aortic arch, descending thoracic aorta, abdominal aorta, carotid arteries, subclavian arteries, axillary arteries, and vertebral arteries) and was calculated as follows: VAMP = (Sum of SUV_{mean} TBRs for all territories + 0.5 if aortic T2-weighted mural signal increased + 0.5 if great vessel T2-weighted mural signal increased) -12. Finally, clinical utility of PET/MRI in LVV was assessed via clinician survey.

2. Please provide separate results for GCA and TAK.

We thank the Reviewer for this suggestion and have performed further analyses to stratify all results by LVV subtype. These new data are presented below (**Figures viii-xii**) and in the revised manuscript (**Supplementary figures 3-7**):

Figure viii. Degree of FDG uptake based on visual comparison with the liver in those with clinically determined active and inactive disease, stratified by LVV subtype. Considering those with active disease, FDG uptake was greater in GCA *versus* TAK in the carotid and subclavian arteries only. Analysis by 2-way ANOVA with Šidák's multiple comparison test.

Figure ix. Degree of FDG uptake based on SUV_{max} TBR analysis in those with GCA and TAK. No difference in FDG uptake was observed between GCA and TAK, including when broken down by disease activity (not shown). Analysis by 2-way ANOVA with Šidák's multiple comparison test.

Figure x. Considering assessment of MR metrics, the only difference observed between GCA and TAK was in mean number of luminal abnormalities, which was greater in TAK. Analysis by unpaired t test.

Figure xi. Considering the longitudinal analyses, numbers were too small, particularly in the TAK group, to draw any meaningful conclusions. Presented are changes in PETVAS (A & B) and cumulative SUV_{mean} TBR (C & D) between baseline and final PET/MRI scans in those with GCA and TAK. The grey box indicates PETVAS cut-off for active *versus* inactive disease. Red dots represent patients with clinically active disease, blue dots represent clinically inactive disease. Analysis by paired t-test

Figure xii. The ability of the VAMP score to distinguish active from inactive disease was stronger in GCA than TAK, though, again, small numbers in the TAK group make conclusions difficult to draw.

These additional analyses highlight some important considerations which we have incorporated into the revised manuscript. To maintain readability, some of this work has been added to the supplementary material. As described above, our opinion is that the role of PET/MRI varies between GCA and TAK; however, our results suggest a role for PET/MRI in both. Future, larger and longer studies may be able to expand upon these differences.

3. Please provide separate results for new-onset and relapsing disease, acknowledging the limited sample size for drawing conclusions within subgroups.

Again, this is an important consideration. We have been responsive to the Reviewer’s request and performed new analyses, presented below, which suggest that there was no statistically significant difference in VAMP score between those with new-onset disease *versus* relapsing disease (**Figure xiii**). However, as noted by the Reviewer, numbers are small, and there is a trend towards a higher score in those with new-onset disease. The manuscript has been updated to acknowledge this.

Figure xiii. Differences in VAMP score between those with new-onset, relapsing, and inactive disease. Analysis by unpaired t test.

4. Elaborate on the characteristics of the four patients with undifferentiated large vessel vasculitis, and explain why you could not classify them as GCA or TAK. Consider the possibility of excluding these patients from calculations.

Diagnostic labelling to LVV subtype were made prior to PET/MRI scanning and based on the opinions of both the referring team and the study investigators. Clinical evaluation together with any previous imaging and blood work informed this decision. Where there was disagreement between the referring team and the investigators, a third, independent, observer was invited to offer a diagnosis, if possible. Despite this, consensus could not be reached in four study subjects. These subjects tended to have an atypical presentation and were of an age at disease onset where either GCA or TAK could be responsible. In two of the four cases, patients presented with evidence of established vascular damage, and it was not possible to say when this damage had occurred, meaning age of onset could not be established. We feel this is reflective of 'real-world' clinical uncertainty, where patients with TAK may present many years or even decades after the true onset of inflammation. Similarly, we have experience of patients with phenotypic GCA who presented at a relatively young age.

Our options were to remove these participants from the analyses, select a diagnosis of either GCA or TAK despite disagreement within the research team, or include them as 'LVV unspecified'. Ultimately, given the modest sample size, we did not want to 'dilute' findings by labelling patients as TAK when in fact they had GCA, or vice versa, and therefore did not see value in applying labels when uncertainty existed. Additionally, we could agree that those labelled as 'LVV unspecified' had a diagnosis of *either* GCA or TAK, and therefore we felt it was important not to exclude them from main analysis.

5. Given the ability of MRI and PET to detect vasculitis in smaller cranial arteries, such as temporal and occipital arteries, consider incorporating them into a scoring system of GCA.

We thank the Reviewer for this excellent point. PET/MRI scanning capabilities and protocols are advancing at an exciting rate. As the Reviewer states, it is now possible to detect vasculitis in the smaller cranial arteries. While we believe this would add value to the VAMP score in those with GCA, our protocol was not designed to evaluate these territories, and so adding these data to the VAMP score within this study is not possible. Future studies should not only include temporal and occipital arteries which may be involved in GCA, but also renal and pulmonary arteries which may be involved in TAK. Such data would hopefully increase the validity of the VAMP score in each of these conditions, and further refine the score. The manuscript has been updated to include this important observation.

6. Your results may have been biased by including patients on treatment at baseline.

Thank you for raising this important point. Our study reflects real-world practice. It is very likely that patients with significant LVV disease activity would be started on some form of therapy prior to baseline PET imaging. Our data suggest that the effect of high dose prednisone started prior to baseline imaging had less of an effect on PET/MRI interpretation (using PETVAS) than on serological measures of systemic inflammation (CRP) (**Figure xiv**).

Figure xiv. PETVAS (left image) and CRP (right image) in those treated with high dose glucocorticoids prior to PET/MR scanning. In those with active disease, exposure to high dose glucocorticoids for >1 week had no apparent effect on PETVAS. Though not significant, the effect of high dose glucocorticoids on CRP in the same cohort appeared more substantial. Analysis by unpaired t-test (PETVAS) and Mann-Whitney test (CRP).

Additionally, in response to the Reviewer’s comments, we have performed new analyses to examine differences between those patients started on *any* immunosuppressive treatment prior to baseline scanning, *versus* those who had received no treatment. This analysis (**Figure xv**) suggests a slight reduction in PET/MRI-based disease activity in those started on treatment. Although these findings are likely to have modest impact on the results overall and the take-home message of the manuscript, we have updated the text to acknowledge this potential source of bias.

Figure xv. The effect of immunosuppressive treatment received prior to baseline PET/MRI scan on PET/MRI metrics (VAMP score). Analysis by unpaired t test.

7. The EULAR recommendations on imaging in large vessel vasculitis have been updated in 2023.

We thank the Reviewer for highlighting this and have updated the manuscript to reflect this.

References

1. Nikuei P, Rajaei M, Roozbeh N, Mohseni F, Poordarvishi F, Azad M and Haidari S. Diagnostic accuracy of sFlt1/PIGF ratio as a marker for preeclampsia. *BMC Pregnancy Childbirth*. 2020;20:80.
2. Levine RJ, Lam C, Qian C, Yu KF, Maynard SE, Sachs BP, Sibai BM, Epstein FH, Romero R, Thadhani R and Karumanchi SA. Soluble endoglin and other circulating antiangiogenic factors in preeclampsia. *N Engl J Med*. 2006;355:992-1005.
3. Girardi G, Yarilin D, Thurman JM, Holers VM and Salmon JE. Complement activation induces dysregulation of angiogenic factors and causes fetal rejection and growth restriction. *J Exp Med*. 2006;203:2165-75.
4. Langer HF, Chung KJ, Orlova VV, Choi EY, Kaul S, Kruhlak MJ, Alatsatianos M, DeAngelis RA, Roche PA, Magotti P, Li X, Economopoulou M, Rafail S, Lambris JD and Chavakis T. Complement-mediated inhibition of neovascularization reveals a point of convergence between innate immunity and angiogenesis. *Blood*. 2010;116:4395-403.
5. Le Roux S, Pepper RJ, Dufay A, Néel M, Meffray E, Lamandé N, Rimbert M, Josien R, Hamidou M, Hourmant M, Cook HT, Charreau B, Larger E, Salama AD and Fakhouri F. Elevated soluble Flt1 inhibits endothelial repair in PR3-ANCA-associated vasculitis. *J Am Soc Nephrol*. 2012;23:155-64.
6. Springer JM, Monach P, Cuthbertson D, Carette S, Khalidi NA, McAlear CA, Pagnoux C, Seo P, Warrington KJ, Ytterberg SR, Hoffman G, Langford C, Hamilton T, Foell D, Vogl T, Holzinger D, Merkel PA, Roth J and Hajj-Ali RA. Serum S100 Proteins as a Marker of Disease Activity in Large Vessel Vasculitis. *J Clin Rheumatol*. 2018;24:393-395.
7. van Sleen Y, Sandovici M, Abdulahad WH, Bijzet J, van der Geest KSM, Boots AMH and Brouwer E. Markers of angiogenesis and macrophage products for predicting disease course and monitoring vascular inflammation in giant cell arteritis. *Rheumatology*. 2019;58:1383-92.
8. Clemente N, Raineri D, Cappellano G, Boggio E, Favero F, Soluri MF, Dianzani C, Comi C, Dianzani U and Chiochetti A. Osteopontin Bridging Innate and Adaptive Immunity in Autoimmune Diseases. *J Immunol Res*. 2016;2016:7675437.
9. Isoda K, Nishikawa K, Kamezawa Y, Yoshida M, Kusuhara M, Moroi M, Tada N and Ohsuzu F. Osteopontin plays an important role in the development of medial thickening and neointimal formation. *Circ Res*. 2002;91:77-82.
10. Prieto-González S, Terrades-García N, Corbera-Bellalta M, Planas-Rigol E, Miyabe C, Alba MA, Ponce A, Tavera-Bahillo I, Murgia G, Espígol-Frigolé G, Marco-Hernández J, Hernández-Rodríguez J, García-Martínez A, Unizony SH and Cid MC. Serum osteopontin: a biomarker of disease activity and predictor of relapsing course in patients with giant cell arteritis. Potential clinical usefulness in tocilizumab-treated patients. *RMD Open*. 2017;3:e000570.

11. Davenport AP, Hyndman KA, Dhaun N, Southan C, Kohan DE, Pollock JS, Pollock DM, Webb DJ and Maguire JJ. Endothelin. *Pharmacol Rev.* 2016;68:357-418.
12. Verhaar MC, Strachan FE, Newby DE, Cruden NL, Koomans HA, Rabelink TJ and Webb DJ. Endothelin-A receptor antagonist-mediated vasodilatation is attenuated by inhibition of nitric oxide synthesis and by endothelin-B receptor blockade. *Circulation.* 1998;97:752-6.
13. Planas-Rigol E, Terrades-Garcia N, Corbera-Bellalta M, Lozano E, Alba MA, Segarra M, Espígol-Frigolé G, Prieto-González S, Hernández-Rodríguez J, Preciado S, Lavilla R and Cid MC. Endothelin-1 promotes vascular smooth muscle cell migration across the artery wall: a mechanism contributing to vascular remodelling and intimal hyperplasia in giant-cell arteritis. *Ann Rheum Dis.* 2017;76:1624-1634.
14. Clemente G, Pereira RMR, Aikawa N, Silva CA, Campos LMA, Alves G, Buchpiguel C, Lima M, Carneiro C, Filho HL, Morbeck F, Neto G, Filho VO, Souza AWD and Terreri MT. Is positron emission tomography/magnetic resonance imaging a reliable tool for detecting vascular activity in treated childhood-onset Takayasu's arteritis? A multicentre study. *Rheumatology.* 2022;61:554-562.
15. Laurent C, Ricard L, Fain O, Buvat I, Adedjouma A, Soussan M and Mekinian A. PET/MRI in large-vessel vasculitis: clinical value for diagnosis and assessment of disease activity. *Sci Rep.* 2019;9:12388.
16. Reichenbach S, Adler S, Bonel H, Cullmann JL, Kuchen S, Bütikofer L, Seitz M and Villiger PM. Magnetic resonance angiography in giant cell arteritis: results of a randomized controlled trial of tocilizumab in giant cell arteritis. *Rheumatology.* 2018;57:982-986.
17. Einspieler I, Thurmel K, Pyka T, Eiber M, Wolfram S, Moog P, Reeps C and Essler M. Imaging large vessel vasculitis with fully integrated PET/MRI: a pilot study. *Eur J Nucl Med Mol Imaging.* 2015;42:1012-24.
18. Stellingwerff MD, Brouwer E, Lensen KDF, Rutgers A, Arends S, van der Geest KSM, Glaudemans A and Slart R. Different Scoring Methods of FDG PET/CT in Giant Cell Arteritis: Need for Standardization. *Medicine.* 2015;94:e1542.

REVIEWERS' COMMENTS

Reviewer #1 (Remarks to the Author):

No further comments

Reviewer #2 (Remarks to the Author):

Thank you for your thorough response. The reviewer remains unconvinced regarding the superiority of the new score over established ones. The study was conducted on a small, heterogeneous cohort comprising both active and inactive patients with giant cell arteritis, Takayasu arteritis, and unclassified large-vessel vasculitis. The score omits the most relevant arteries for giant cell arteritis. The inclusion of patients already undergoing treatment at baseline likely introduces bias into the results.

Reviewer #3 (Remarks to the Author):

Re: response to the reviewer-2 comments

All the concerns/comments raised by the reviewer have been addressed satisfactorily.